



# Geographic-information-system-based topographic reconstruction and geomechanical modelling of the Köfels rockslide

**Christian Zangerl**[1], **Annemarie Schneeberger**[1,2], **Georg Steiner**[1,3], **and Martin Mergili**[1,4]

[1]Department of Civil Engineering and Natural Hazards, Institute of Applied Geology, University of Natural Resources and Life Sciences (BOKU), Vienna, 1190, Austria
[2]Institute of Geography, University of Innsbruck, Innsbruck, 6020, Austria
[3]Amt der Kärntner Landesregierung, Klagenfurt, 9021, Austria
[4]Department of Geography and Regional Science, University of Graz, Graz, 8010, Austria

**Correspondence:** Christian Zangerl (christian.j.zangerl@boku.ac.at)

**Abstract.** The Köfels rockslide in the Ötztal Valley (Tyrol, Austria) represents the largest known extremely rapid landslide in metamorphic rock masses in the Alps. Although many hypotheses for the trigger were discussed in the past, until now no scientifically proven trigger factor has been identified. This study provides new data about the (i) prefailure and failure topography, (ii) failure volume and porosity of the sliding mass, and (iii) numerical models on initial deformation and failure mechanism, as well as shear strength properties of the basal shear zone obtained by back-calculations. Geographic information system (GIS) methods were used to reconstruct the slope topographies before, during and after the event. Comparing the resulting digital terrain models leads to volume estimates of the failure and deposition masses of 3100 and 4000 million $m^3$, respectively, and a sliding mass porosity of 26 %. For the 2D numerical investigation the distinct element method was applied to study the geomechanical characteristics of the initial failure process (i.e. model runs without a basal shear zone) and to determine the shear strength properties of the reconstructed basal shear zone. Based on numerous model runs by varying the block and joint input parameters, the failure process of the rock slope could be plausibly reconstructed; however, the exact geometry of the rockslide, especially in view of thickness, could not be fully reproduced. Our results suggest that both failure of rock blocks and shearing along dipping joints moderately to the east were responsible for the formation or the rockslide. The progressive failure process may have taken place by fracturing and loosening of the rock

mass, advancing from shallow to deep-seated zones, especially by the development of internal shear zones, as well as localized domains of increased block failure. The simulations further highlighted the importance of considering the dominant structural features of the rock mass. Considering back-calculations of the strength properties, i.e. the friction angle of the basal shear zone, the results indicated that under no groundwater flow conditions, an exceptionally low friction angle of 21 to 24° or below is required to promote failure, depending on how much internal shearing of the sliding mass is allowed. Model runs considering groundwater flow resulted in approximately 6° higher back-calculated critical friction angles ranging from 27 to 30°. Such low friction angles of the basal failure zone are unexpected from a rock mechanical perspective for this strong rock, and groundwater flow, even if high water pressures are assumed, may not be able to trigger this rockslide. In addition, the rock mass properties needed to induce failure in the model runs if no basal shear zone was implemented are significantly lower than those which would be obtained by classical rock mechanical considerations. Additional conditioning and triggering factors such as the impact of earthquakes acting as precursors for progressive rock mass weakening may have been involved in causing this gigantic rockslide.

**Published by Copernicus Publications on behalf of the European Geosciences Union.**

## 1   Introduction

In mountain areas, life and property are often put at risk by landslide processes (e.g., Dai et al., 2002; Nadim et al., 2006; Margottini et al., 2013; Sassa et al., 2014). Rapid collapses of huge mountain slopes – and resulting process chains – have repeatedly evolved into catastrophic events (e.g., Evans and DeGraff, 2002; Govi et al., 2002; Genevois and Ghirotti, 2005; Evans et al., 2009a, b). An adequate understanding of the mechanisms of the initial failure and extremely rapid movement processes is one key for the implementation of effective risk reduction strategies. The analysis of past – even fossil – events may contribute to a better understanding of landslide processes and therefore help us to develop and to improve methods for hazard and risk mitigation (Kilburn and Pasuto, 2003).

Known as the largest landslide in metamorphic rock throughout the European Alps, the Köfels rockslide represents such a fossil landslide (see Sect. 2 for a detailed description). In contrast to numerous deep-seated rockslides in foliated metamorphic rocks characterized by movement rates of a few centimetres to decimetres per year and without indications of total slope failure (Zangerl et al., 2015), the Köfels rockslide is a prominent case study for a sudden slope failure with extremely rapid movement velocities. This can be clearly demonstrated by the occurrence of frictionites which were found at outcrops on the deposited sliding mass (Erismann et al., 1977). Even though this giant landslide has been the subject of numerous studies focussing on the genesis of the frictionites, age of the event, spatial distribution of the source area, volume of the rockslide mass and geomechanical aspects concerning the trigger and failure mechanisms (e.g., Pichler, 1863; Milton, 1964; Preuss, 1974, 1986; Erismann et al., 1977; Preuss et al., 1987; Erismann and Abele, 2001; Brückl et al., 2001, 2010; Brückl and Parotidis, 2001, 2005; von Poschinger, 2002; Sørensen and Bauer, 2003; Prager et al., 2009, Nicolussi et al., 2015), the conditioning and triggering factors of the Köfels rockslide still remain unknown and speculative.

Computer models focussing on the rockslide geometry and geomechanical processes may help to increase our understanding of the mechanisms of rock slope failure. Although models are always a rough simplification of reality, some are useful to explore specific aspects such as initial failure processes, slope deformations, rockslide volumes or critical values of geomechanical parameters at failure. In the context of this study two types of models, i.e. topographical and geomechanical models, are relevant. Brückl et al. (2001) were the first ones who reconstructed the 3D pre-failure topography and failure geometry of the Köfels rockslide on the basis of seismic measurements and terrain models, and they derived parameters such as failure and deposition volumes, porosity, the initial and average sliding angles, and the release of potential energy.

In our study we used new high-resolution (1 m raster data) ALS-based (airborne laser scanning) digital terrain models, new geological mapping data and pre-existing data from seismic measurements to re-build and re-analyse the pre- and post-failure topographies and geometries of the rockslide. Based on this topographic reconstruction by using geographic information system (GIS) analysis methods, a geometrical and kinematical rockslide model was developed. Conclusions can be made about the failed and deposited volumes and consequently the change of rock mass porosity induced by the rapid sliding and fracturing and loosening processes.

Concerning geomechanics of the rockslide at initial failure state and movement, several attempts were made to investigate the mechanisms and to back-calculate rock mass properties. Erismann et al. (1977) developed a kinematic and thermodynamic model to explain the energy release necessary for the formation of the frictionites that were found at the Köfels site (see Sect. 2). Brückl and Parotidis (2001) set up a 2D elastic and elasto-plastic continuum model to estimate the geomechanical rock mass properties of the Köfels rockslide. In their approach they applied the 2D finite element method to explore the initial phase of the failure process by studying the creeping and strength degradation of the rock mass. The model suggests that the Köfels rockslide was formed due to the progressively weakening strength of the rock mass, which was initiated at the foot of the slope and propagated uphill. Furthermore, the model calculations determined surprisingly low friction angles of the rock mass, ranging between 20 and 24°, to induce slope failure. In another approach, Brückl and Parotidis (2005) proposed a model with focus on time-dependent strength degradation and slope failure under low stress regimes such as rock mass creep and subcritical crack growth. They suggest that subcritical crack growth is a primary geomechanical process which, after glacier retreat, is able to explain the considerable rock mass strength weakening needed for failure.

However, the extraordinary low-strength properties of the rock mass that were back-calculated by 2D continuum approaches for the failure state raise questions:

- – Can we plausibly reconstruct the topography to provide a realistic pre-failure topography for the geomechanical modelling?

- – How could the initial failure and slope deformation process have taken place?

- – How can the strength of such a strong granitic rock mass reduce to such small values needed to promote failure?

- – Are there any structural particularities in the Köfels rockslide area that may have contributed to slope failure and what is the influence of the pre-existing fracture network?

– Why do we observe only one such giant and extremely rapid rockslide characterized by a flat to moderately dipping failure surface in the Ötztal–Stubai crystalline basement?

Given that, so far, only 2D continuum models have been applied to investigate the failure mechanisms of the Köfels rockslide, we believe that, though representing a valid approach, additional types of models, e.g. discontinuum models, are useful to adequately capture the complexity of the phenomenon. Discontinuum models such as the distinct element method have the advantage that the geometry of the rockslide mass and the discrete basal shear zone can be implemented directly based on geometrical and structural field observations and GIS reconstructions. Geomechanically, the basal shear zone, i.e. stepped rupture surface, can be considered in the model as a discrete narrow zone. In order to fill this gap, we set up a 2D discontinuum model of the Köfels rockslide based on the geometry obtained by the topographic reconstruction and by applying the Universal Distinct Element Code (UDEC; Itasca, 2020). The initial failure process was studied by considering the main structural characteristics based on geological field surveys. The aim was to investigate how the rockslide geometry and the basal shear surface (zone) was formed during the initial failure process. In addition, back-calculations of the critical angle of friction along the basal shear zone assuming no groundwater flow conditions and groundwater flow are conducted under quasi-static conditions. These back-calculations were done to determine the shear strength properties, i.e. friction angle and cohesion, of the predefined and field-based basal shear zone needed to promote failure. The models were performed to explore the influence of fracture water pressure in the rock mass and basal shear zone resulting from high groundwater levels for provoking this giant landslide.

The numerical modelling study was supplemented by a geological field survey searching for instability-relevant discontinuities of different origin and scale. This was done to investigate the impact of discontinuities which ideally are dipping moderately towards the east, acting as weakness zones and thus reducing the overall rock mass strength. Particular focus was given to the identification of low-strength brittle fault zones composed of gouges and breccia characterized by a high persistence.

Next, we introduce the study area, the Köfels rockslide (Sect. 2). Then, we explain the methods applied for the topographic reconstruction and geomechanical modelling (Sect. 3). We present (Sect. 4) and discuss (Sect. 5) the results before concluding with the key messages of this study (Sect. 6).

## 2 Study area and data

### 2.1 Geographic and geologic setting

The Köfels rockslide (Figs. 1 and 2) occurred in the central part of the north–south striking Ötztal Valley (Tyrol, Austria), at present at an elevation between 950 and 1100 m a.s.l. Surrounded by up to 3000 m high summits, this area is deeply incised in the poly-metamorphic Ötztal complex, a major thrust unit belonging to the Upper Austro-alpine basement nappes (Prager et al., 2009). Lithologically, different types of metamorphic rocks, i.e. paragneisses, quartzites and mica schists, with intercalations of orthogneisses, amphibolites and eclogites are encountered (Hammer, 1929; Purtscheller, 1978). The complex ductile and brittle structural setting results from polyphase and heteroaxially deformations and is attributed to at least three distinct orogeneses and their corresponding regional metamorphic overprint. In contrast to numerous petrological and geochronological studies, the brittle deformation history and the related structures of the Ötztal basement have not been studied so far in detail but would be highly relevant for geomechanical purposes. However, Prager et al. (2009) provide some data concerning the discontinuity network in the surroundings of the Köfels rockslide.

During the Quaternary period, the Ötztal Valley was influenced by repeated glacier fluctuations causing valley incision and glacial and fluvial erosion, as well as sediment accumulation. Valley deepening and steepening lead to substantial stress redistributions in the rock slopes, which in turn initiates time-dependent progressive failure processes in the fractured rock mass and may expose preferentially orientated failure surfaces.

### 2.2 The Köfels rockslide

The age of the Köfels rockslide was determined several times through radiocarbon dating of wood buried by the rockslide deposits (Ivy-Ochs et al., 1998), surface exposure dating of rockslide boulders (Kubik et al., 1998) and actually by tree-ring analysis and radiocarbon dating of new wood samples (Nicolussi et al., 2015). The last dating campaign, yielding 9527–9498 cal BP, led to a significant refining of the timing of the Köfels landslide event and even was able to constrain the season during which the event occurred.

The main source of the slide is located in competent fractured orthogneisses (augengneiss) around the small village of Köfels. Only the southern head scarp area of the rockslide is composed of paragneissic rock. The head scarp located at the western slope of the central Ötztal Valley is very steep with inclinations of up to 40–80°. Comprising a failure volume of more than 3000 million m$^3$ the Köfels rockslide demonstrates a particular event of very rapid large-scale failure in metamorphic rock mass (Brückl et al., 2001). Typically, such rapid rockslides characterized by a moderately inclined basal failure surface occur in carbonatic rock masses (Prager et

https://doi.org/10.5194/nhess-21-1-2021

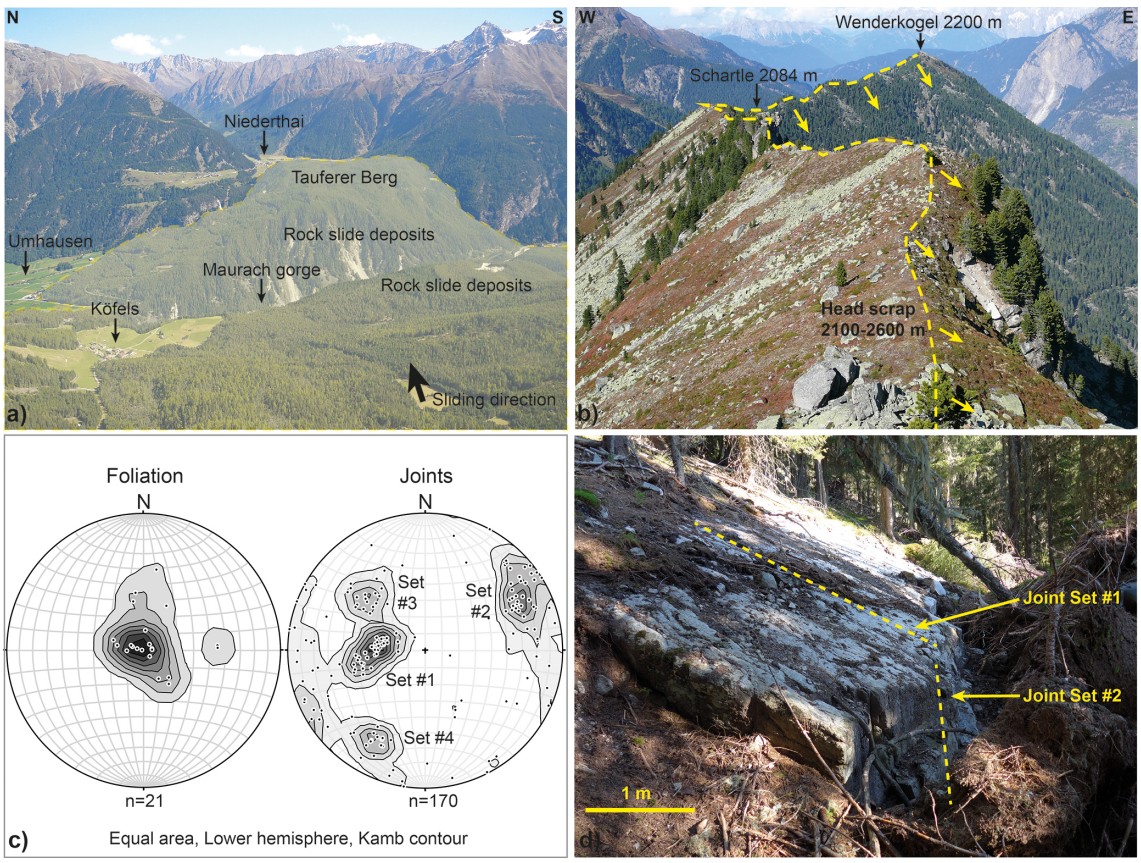

**Figure 1. (a)** Panoramic view of the Köfels rockslide deposits from head scarp towards the east with the Maurach Gorge cutting through the deposits (centre) and the backwater sediments in Niederthai (right), **(b)** view of the head scarp from south to north, **(c)** measured foliation and joint planes (poles to planes) in the surroundings of the central part of the head scarp, and **(d)** outcropping rupture surface formed along a moderately dipping plane of joint set #1 and linked with joint set #2 (stepped failure plane). Stereonet pole and contour plots were created using the Stereonet software (Allmendinger, 2018; Cardozo and Allmendinger, 2013).

al., 2008). The displacement of the sliding rock mass was initiated at the east-facing slope south of Wenderkogel (see Figs. 1 and 2) and stopped at the opposite slope in the east at the entrance of the tributary valley, the Horlachtal Valley, where it collided with massive bedrock. The centre of mass displaced by around 2.6 km (Sørensen and Bauer, 2003), reaching a velocity of approximately 50 m/s (Erismann et al., 1977). The main rockslide deposit blocked the Ötztal Valley and formed a prominent valley spur of fractured and disintegrated orthogneiss. Erismann and Abele (2001) proposed that the mass was split into two parts with the lower one arresting due to the collision within the steep valley slope and the upper one which continued its movement, thus creating an additional internal sub-horizontal shear zone. The Tauferer Berg (see Figs. 1 and 2) was formed when the upper mass continued its movement towards the Horlachtal Valley for approximately 1 more kilometre and ran up for approximately 100 m. Though plausible, evidence for a distinct internal shear zone was claimed by Preuss (1986), but proof for the existence of such a feature has not yet been

found in the field. It seems even more plausible that the immense internal rock mass deformation during the movement and the adaptation to the terrain surface were based on the formation of numerous internal shear zones. The disintegration of the rock mass during the slide event caused a very heterogeneous highly fractured and partly crushed rock mass, with shear zones composed of gouges and breccias and zones with blocks of more than 10 m diameter (Sørensen and Bauer, 2003). Furthermore, zones that are characterized by high fracture frequencies only marginally increased in comparison to those commonly observed in undisturbed fractured rock masses. This distinctive fragmentation of rock led to radon gas emissions and locally radioactive springs which affected the population in Umhausen and caused noticeably high cancer rates (Purtscheller et al., 1995).

After the slide event, a temporary lake flooded the basin of Längenfeld, impounded by the valley spur (Ampferer, 1939). As a result of the flooding backwater sediments were deposited in the basin of Längenfeld, as well as in the blocked tributary of the Horlachtal Valley at Niederthai. Ac-

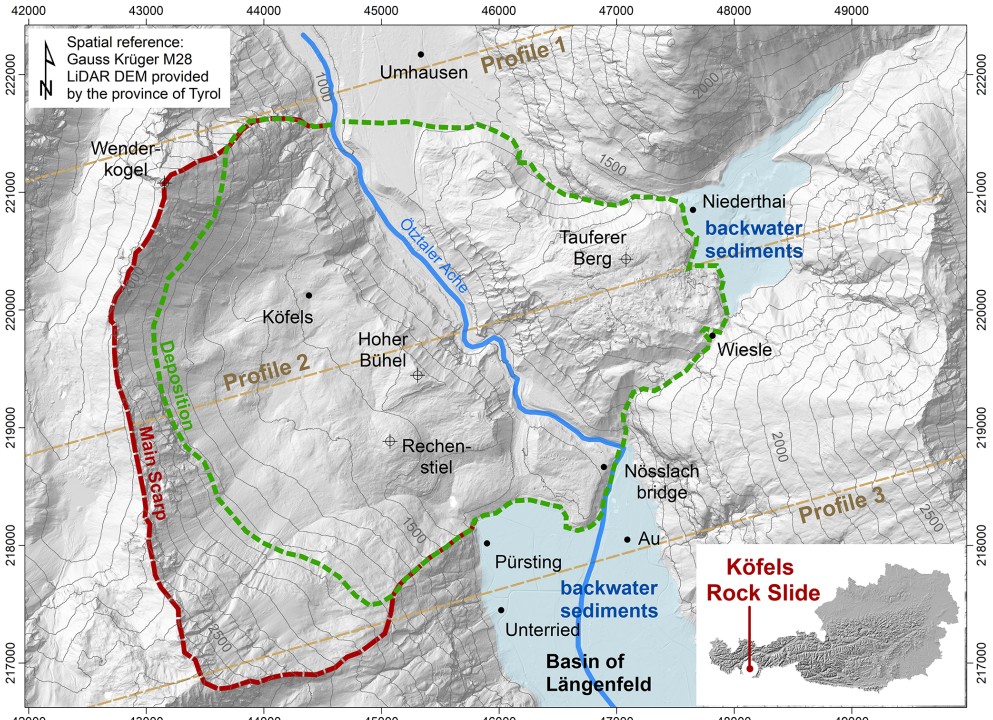

**Figure 2.** Overview map of the Köfels rockslide area.

cording to drilling data from von Klebelsberg (1951) and Ampferer (1939), the lacustrine sediments reach a maximum thickness of 92 m. Later on, the river Ötztaler Ache cut into the rockslide deposits, forming the Maurach gorge by fluvial erosion (see Figs. 1 and 2; Erismann and Abele, 2001).

When the mountain slope collapsed, an amount of about $1.65 \times 10^7$ GJ of energy was released. This value was estimated by Erismann and Abele (2001) with respect to volume, density and vertical displacement of the rock mass. The high amount of released energy led to partial melting of the orthogneissic rock at the progressively exposed sliding surface but also around internal shear zones and the development of a fused rock (i.e. pumice, frictionites, hyalomylonites), the presence of which was interpreted in various ways over the years (e.g., Pichler, 1863; Preuss, 1974; Erismann et al., 1977; Masch et al., 1985; Weidinger et al., 2014).

Though subject of research for more than one century, the question of the causes and maybe the "single" trigger for the Köfels rockslide remains still open. Most probably a combination of various conditioning and interacting triggering factors led to the release of this giant slide.

Given that the collapse of Köfels occurred several thousand years after valley deglaciation, time-dependent progressive failure processes such as sub-critical crack growth and fracture propagation were caused by over-steepening of the valley flanks which is assumed to have provoked unstable conditions in the slope. This long-term disintegration of rock is seen as a prerequisite for the development of a large-scale

rockslide (Prager et al., 2009; Brückl and Parotidis, 2005; Abele, 1994). Moreover, permafrost degradation is suspected to have influenced the failure of many Holocene deep-seated rockslides (Abele, 1994) – a phenomenon that gains new relevance considering the degrading permafrost in today's mountains influenced by modern climatic changes (e.g. Gruber and Haeberli, 2007; Huggel et al., 2012). Abele (1994) and Weidinger (2006) describe active tectonics, i.e. earthquakes, as one main background condition provoking large rockslide events due to dynamic loading. Considering the present low seismic activity in the Ötztal Valley, Sørensen and Bauer (2003) question an earthquake as a possible trigger for the event.

## 2.3 Data

An up-to-date digital elevation model (DEM), gained by airborne laser scanning (lidar), of the investigated area was obtained from the governmental service for maps of Tyrol, TIRIS, at a spatial resolution of 1 m. Topographic and geologic information on the situation before and after the Köfels rockslide are given through studies of von Klebelsberg (1951), Brückl (2001), Heuberger (1994), and Prager et al. (2009). Data from several boreholes from von Klebelsberg (1951) were used in this work. Additionally, reflection and refraction seismic measurements were conducted between 1986 and 1990 (Brückl and Heuberger, 1993; Brückl et al., 1998, 2001). In the framework of a hydroelectric power project an investigation drift was drilled into the Tauferer

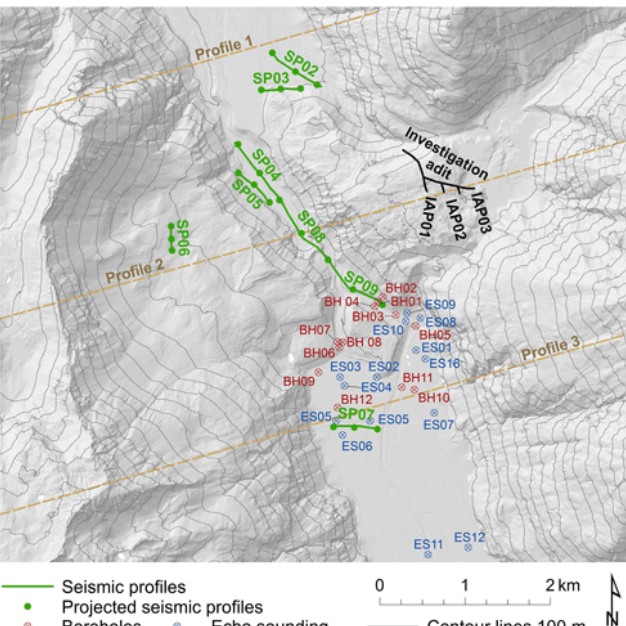

**Figure 3.** Borehole data (BH), echo soundings (ESs) and seismic profiles (SPs) used for the topographic reconstruction of the Köfels rockslide.

Berg in 1952 which provides additional information about the geological setting of the site (Brückl et al., 2001; Ascher, 1952). Figure 3 provides an overview of the geophysical and drilling data used for the study.

## 3 Methods

### 3.1 Reconstruction of rockslide topography, volume and porosity

Three topographic profiles were constructed based on the drilling and seismic data provided by von Klebelsberg (1951), Heuberger (1994), and Brückl et al. (2001): Profile 1 is set north of the rockslide zone through the basin of Umhausen, Profile 2 lies within the sliding surface, and Profile 3 is south of the rockslide zone in the basin of Längenfeld (see Figs. 2 and 3).

Figures 2 and 3 show the location of the WSW–ENE profiles 1 and 3. The two profiles display the pre-failure topography reconstructed from the seismic and borehole data and the up-to-date situation. The seismic profiles were projected to the topographic sections and transformed into point data used as input for the GIS-based topographic reconstruction. All spatial analysis tasks were performed using the ArcGIS software (Esri, 2014).

For the reconstruction of the past topographic scenarios, an intermediate horizon of the reflection seismic data was assumed as the top of compacted sediments made up of an old valley infill, which was interpreted to be older than the

Köfels rockslide. These sediments were buried by the rockslide mass and their upper boundary used for the reconstruction of the topographic scenario in the valley as it was before the Köfels rockslide. The deepest horizon with a maximum depth of 400 m was interpreted as the compact rock surface – identical with the sliding plane of the rockslide at the flanks of the valley (Brückl et al., 2001).

The available data are then used to three-dimensionally reconstruct four topographic situations, assuming a U-shaped pre-failure valley topography, as well as a curved failure surface:

1. the pre-failure topography before the Köfels rockslide event and before the alluvium north and south of the site was deposited

2. the topography of the failure surface with the deposits completely removed from the model to illustrate the basal shear zone and without the alluvial deposits north and south of the rockslide

3. the post-failure topography without the alluvial deposits and with the Köfels rockslide deposit in the valley before the incision by the Ötztaler Ache

4. the up-to-date topography where the Maurach gorge was created by the incision of the Ötztaler Ache into the deposits and the alluvium deposited in the basins of Längenfeld and Umhausen (see Fig. 2).

Within the rockslide mass only information from seismic profiles and the investigation adit was given. In the failure area only a little data were available. The reconstruction of the pre-failure topography of the Köfels rockslide was built on the contour lines of the escarpment of the up-to-date DEM. The hypothetic pre-failure slope between the edges of the escarpment was assumed plane. This simple way of reconstruction does not require additional assumptions not supported by observations.

The failure and the deposition volumes of the Köfels rockslide mass were computed from the three reconstructed DEMs:

$$V_F = \sum_{i=1}^{i=m} A \left( z_{i,1} - z_{i,2} \right), \tag{1}$$

$$V_D = \sum_{i=1}^{i=m} A \left( z_{i,3} - z_{i,2} \right), \tag{2}$$

where $V_F$ and $V_D$ are the failure and deposition volumes, and $z_{i,1}$, $z_{i,2}$ and $z_{i,3}$ represent the reconstructed elevation of the pixel $i$, the numbers referring to the stages given above. $A$ is the area of one pixel, and $m$ is the number of pixels.

Based on the results of the volumetric calculation in ArcGIS, the porosity of the rock mass before and after failure of the Köfels rockslide was estimated. Porosity is defined as

the ratio of void space to the total volume of soil or rock (Fetter, 2001):

$$n = \frac{V_p}{V_s + V_p},$$ (3)

where $n$ is the porosity, $V_p$ is the volume of void space, and $V_s$ is the volume of solids.

## 3.2 Discontinuity mapping and rock mass characterization

In order to characterize the discontinuity network and the rock mass strength a field survey based on outcrop and scanline mapping was performed at the slopes of the head scarp. Particular focus was given to detect brittle fault zones composed of gouge and breccia, which are dipping moderately towards the east and therefore could have acted, at least partly, as the basal shear zone of the Köfels rockslide. Estimation of rock mass strength and shear strength of discontinuities was done based on field surveys for rock mass characterization, application of empirical methods (Hoek and Brown, 1997) and analyses of existing laboratory tests to provide data for the comparison with results obtained by the numerical modelling study.

## 3.3 Distinct element modelling of the Köfels rockslide

### 3.3.1 Modelling strategy

A discontinuum model (i.e. distinct or discrete element method) has not yet been applied for geomechanical modelling of the Köfels rockslide (see Sect. 1). The major advantage of discontinuum modelling compared to a classical continuum approach is that (i) the structural anisotropy of the rock mass caused by a discrete fracture network can be considered and (ii), typical for slides, a distinct, field-based and, in relationship to the model size, thin basal sliding zone which is able to accumulate large shear displacements can be implemented and modelled.

In this study we performed more than 50 model runs and established two types of distinct element model scenarios (scenarios I and II). Model scenario I was built to investigate the initial failure and deformation mechanism of the rockslide, primarily characterized by internal deformation of the slope and the development of the fully persistent basal rupture surface (shear zone). To achieve this, the model considered the main characteristics of the in situ fracture network (i.e. shear and opening displacements), as well as rock block deformation and failure (i.e. the Mohr–Coulomb constitutive model). However, a basal shear zone was not implemented. Model scenario II was built to perform back-calculations of the shear strength properties of the basal shear zone at failure state. Both scenarios were calculated either by considering dry conditions or groundwater flow by applying a water pressure in the discontinuities. The aim of this modelling campaign was

a  to study the initial failure mechanism before the basal rupture zone was formed

b  to assess the impact of the pre-existing fracture network on the failure geometry of the rockslide

c  to investigate the role of discontinuity and rock block properties and constitutive relationships

d  to back-calculate the shear strength properties of the basal shear zone at the failure state and its bandwidth under the framework of the comprehensively reconstructed pre-failure topography

e  to determine internal deformation behaviour of the rockslide mass and the influence of the shape of the basal shear zone on it

f  to draw conclusions on the role of water pressure as a possible trigger mechanism of the event by evaluating the required shear strength properties against typical values of fractured rock masses.

For the numerical study 2D distinct element models based on the code UDEC (Itasca, 2020) were designed. This software tool characterizes a discontinuous rock mass by an assembly of discrete blocks with contacts or interfaces in between. A continuum mesh of finite-difference zones provides the deformability of the blocks according to elastic or elastoplastic constitutive models. During the calculation procedure the deformable blocks interact mechanically at their surfaces and corners. Block velocities and displacements are determined, with the calculation procedure being repeated until a balanced state of equilibrium or ongoing failure is reached.

## 4 Results

### 4.1 Reconstruction of rockslide topography, volume and porosity

Figures 5 and 6 illustrates the results of the three-stage topographic reconstruction of the Köfels rockslide. Whilst stage 2 represents a theoretical situation that has never occurred in this way (however, it is necessary to reconstruct the rockslide volumes; see Eqs. (1) and (2)), stages 1 and 3 represent hypothetic morphologies directly before and after the event. Note that the very smooth pre-failure topography of the failure area most probably does not resemble the original shape of the mountain slope before failure (see Figs. 5 and 6a); however, given the fact that there are no data supporting more advanced reconstruction methods, we considered this approach a reasonable approximation. Stage 4 represents the situation observed today. Comparing Fig. 6c and d indicate that those morphologic processes have shaped the site since the event, most significantly the incision of the Maurach gorge by the Ötztaler Ache River into the rockslide deposits and the deposition of lake sediments in the basins of Umhausen and

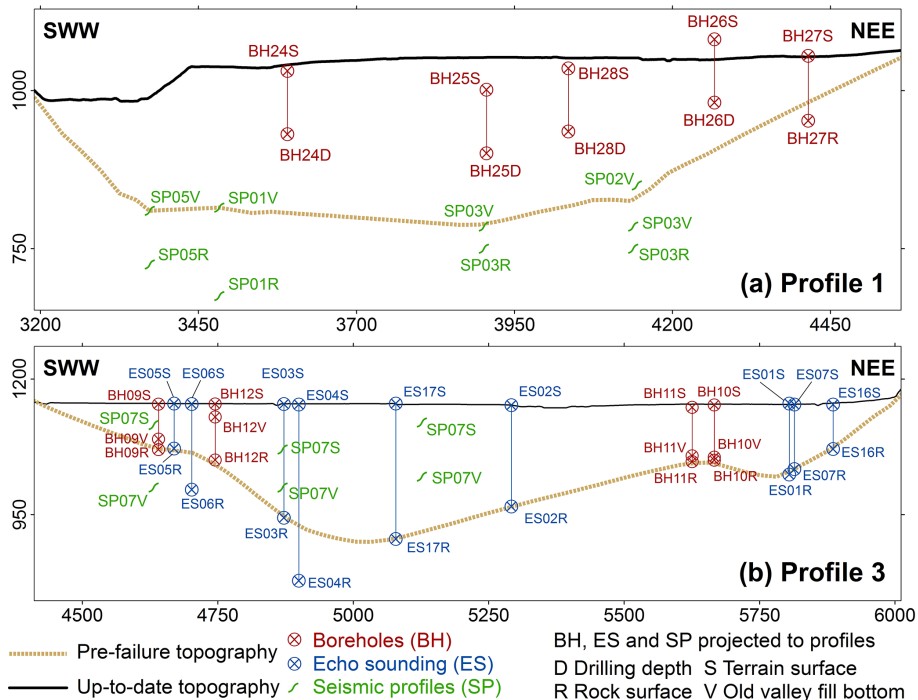

**Figure 4.** Profiles through the valley bottom in the Köfels rockslide area (see Figs. 2 and 3). **(a)** Profile 1 in the north of the Köfels site in the basin of Umhausen. **(b)** Profile 3 in the south of the Köfels site in the basin of Längenfeld. Note that the point data and seismic profiles (see Fig. 3) are projected to the profile planes and therefore do not necessarily correspond to the topographic surfaces shown.

Längenfeld, as well as the Horlachtal Valley (see Figs. 1, 2 and 5).

Applying Eqs. (1) and (2) to the reconstructed topographies, a failure volume of $V_F = 3100$ million m$^3$ and a deposition volume of $V_D = 4000$ million m$^3$ were obtained (Fig. 7). Based on these volumetric reconstructions of failure and deposition masses, considerations about the porosity before and after the Köfels rockslide were made. Typical porosities for intact granitic rocks caused by microfractures are around 1 %–2 %, not considering any mesoscale joints (Zangerl et al., 2003). Taking into account joints in the rock mass the porosity increase to 2 %–5 % (Fetter, 2001). Assuming a pre-failure porosity of the fractured granitic rock mass of 5 % and a constant volume of the solid content of the rock mass $V_s$ before and after the collapse of the mountain slope, Eq. (3) predicts a porosity of the fractured rock mass after the sliding event of approximately 26 %. Consequently, we estimated an increase in the mean porosity from a few percentage points to 26 % resulting from the disintegration of the rock mass during the Köfels rockslide.

## 4.2   Discontinuity and rock mass characterization

The overall structural setting of the Köfels rockslide scarp area has already been comprehensively described in Prager et al. (2009). Nevertheless, in this study new discontinuity data were obtained during an outcrop mapping campaign in

the orthogneissic rocks around the central part of the head scarp. Data comprising discontinuity orientation, frequency, spacing, length, roughness and strength were sampled by scanline and outcrop surveys to determine the structural anisotropy and to estimate roughly the strength properties of the rock mass. The orthogneissic rock is foliated, therefore highly anisotropic with a mean dip direction and dip angle of 114/07 (Fig. 1c). At mesoscale, the rock mass is fractured by four joint sets. One primary joint set, labelled as set #1, is dipping moderately towards the east and varying by around a mean dip direction/dip angle of 090/32 (Fig. 1c). Joints assigned to set #1 are dipping sub-parallel to the exposed scarp surface and thus are part of the basal shear zone. Remarkably, these joints feature a medium to very high persistence, reaching lengths of several tens of metres and a surface roughness defined as rough and stepped (ISRM 1978). According to the approach of Barton and Choubey (1977) a mean joint roughness coefficient of JRC = 10 was determined. Occasionally, some surfaces of fractures orientated sub-parallel to set #1 are coated with quartz minerals, representing vein fillings which were most likely sheared and exposed during the rockslide events. The appearance of striations on these fractures suggest a tectonic origin, i.e. shear fractures or fault planes. A further dominant joint set (#2) is dipping steeply towards west-southwest (dip direction/dip angle of 242/70; Fig. 1c). However, in some areas surrounding the head scarp, set #2 dips steeply towards the east (Fig. 1c). The stepped topogra-

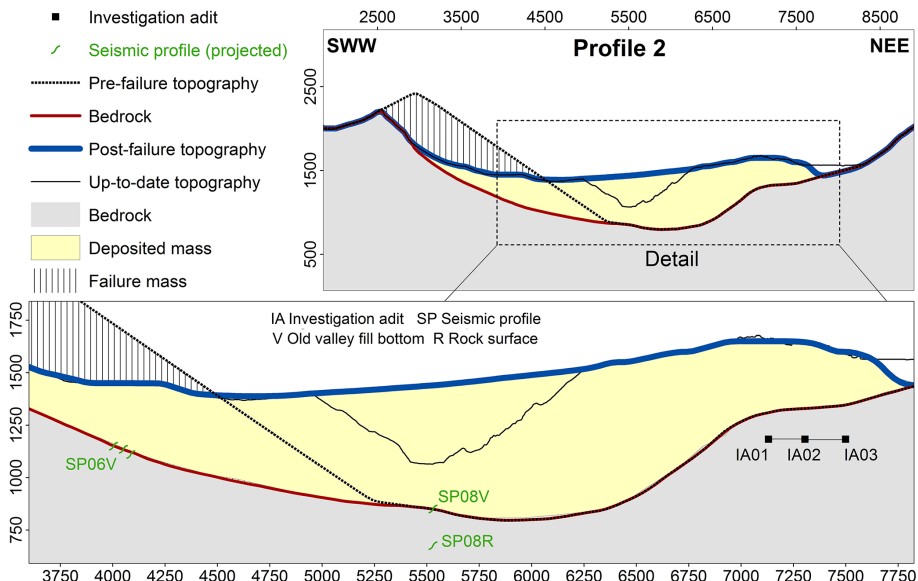

**Figure 5.** Profile 2 (see Figs. 2 and 3) through the Köfels site with the three reconstructed stages and the up-to-date topography.

phy of the scarp flank observed in the upper scarp area originated by the intersection of these two joint sets, forming a stepped failure surface (Fig. 1d). In addition, two less prominent joint sets, i.e. set #3 clustering around a mean of 133/47 and overlapping with set #1, as well as set #4 with a mean of 030/65, were measured (Fig. 1c).

Based on field measurements in the orthogneissic rock mass a mean total joint spacing of around 0.6 m and a mean block size of approximately $V_b = 0.3\,\text{m}^3$ were obtained. Special attention was given during the field campaigns to detect brittle fault zones of tectonic origin with a preferable orientation, dipping moderately to the east and with fault zone infillings of gouge and breccia. These brittle fault zones, if available, could have acted as low-strength weakness zones and therefore been responsible to a certain degree for the rockslide formation. Although a detailed exploration of the terrain was carried out, no such structures could be found. In addition, the spatial analysis of high-resolution lidar-based digital elevation models (DEMs; 1 m raster) also provide no evidence for such brittle faults. On the contrary, most brittle fault zones mapped are inclined steeply and are striking west-northwest–east-southeast (major set), east-northeast–west-southwest (minor) and north-northwest–south-southeast. As already mentioned above, only mesoscale fractures coated with striations were found in the scarp area, representing structures with shear markers. Based on the current level of knowledge there is no clear evidence that low-strength brittle fault zones were involved as part of the basal rupture surface in the initial progressive failure process of the rockslide.

In order to assess the strength of the rock mass, the geological strength index (GSI) characterization method proposed by Cai et al. (2004) and Hoek and Brown (1997) was applied. From the field survey a GSI $\geq$ 55 and from block size/joint spacing a GSI $= 55$ were obtained for the orthogneissic rock. Uniaxial compressive strength (UCS) tests performed on orthogneisses show a mean UCS value of 125 MPa (nine tests were performed on similar rocks in the context of a dam project). In order to consider the influence of long-term loading on the strength of the intact rock (rock creeping, subcritical crack growth), the uniaxial compressive strength is reduced to 40 % of the test results, which yields 50 MPa (Damjanac and Fairhurst, 2010). In order to assess the lower limit of the rock mass strength, the GSI was further reduced to 45 by assuming an intact rock parameter $m_i = 15$ (Hoek and Brown, 1997). Based on these parameters and the Hoek–Brown failure criterion, a rock mass shear strength of $c_{\text{rm}} = 2\,\text{MPa}$ and $\varphi_{\text{rm}} = 35°$ was estimated. The intact rock shear strength of orthogneissic rock was determined by triaxial laboratory testing and obviously is much higher in the range of $c_i = 16$ to 41 MPa and $\varphi_i = 31$ to 40° (tests were performed on similar rocks in the context of a dam project). The shear strength of the joints could not be measured in situ and was therefore estimated based on the Barton's empirical approach (Barton and Choubey, 1977). The shear strength of unfilled joints is influenced by the roughness, the strength of the joint surface and the normal stress acting on the discontinuity. On the basis of geometrical considerations and modelling results, it was assumed that the in situ normal stresses acting on the basal shear zone were in the range between 4 and 18 MPa. According to the method of Barton and Choubey (1977) a friction angle ranging from 32 to 35°, by neglecting cohesion ($c = 0\,\text{MPa}$), was roughly estimated.

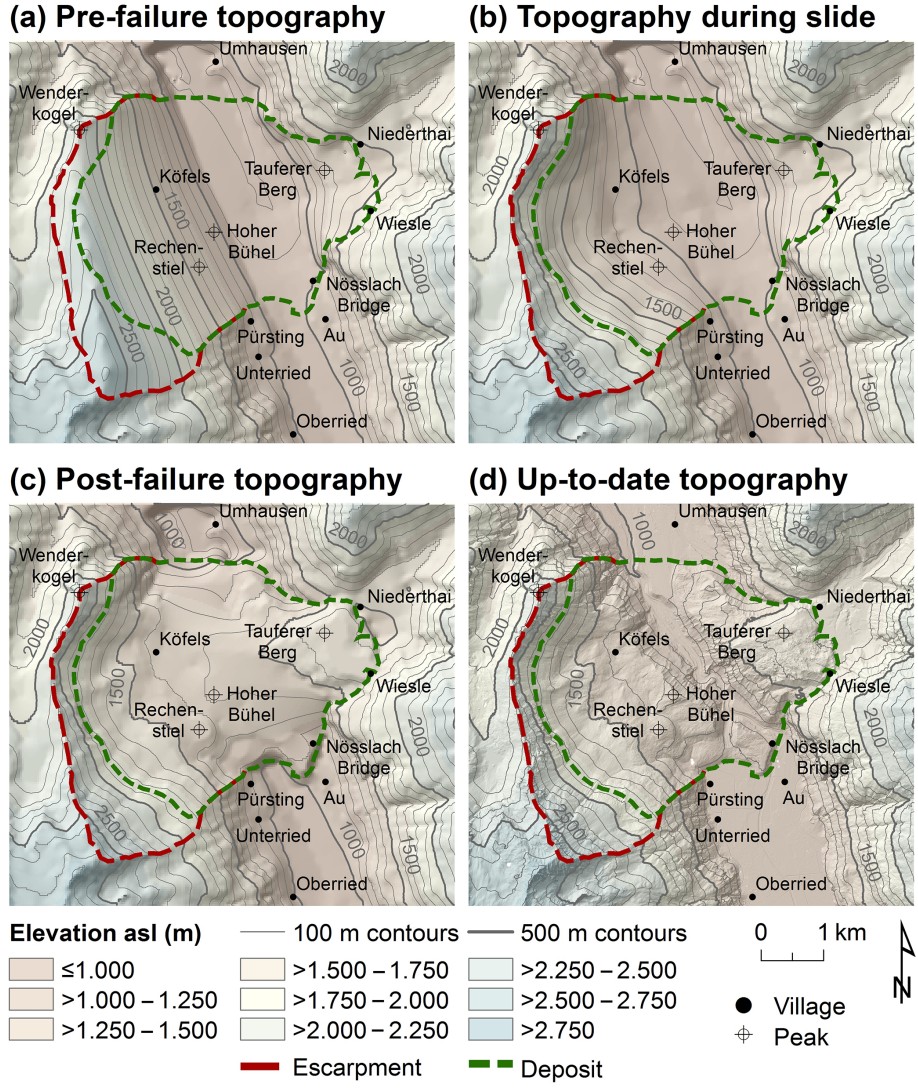

**Figure 6.** DEM of the three reconstructed stages and the up-to-date topography: **(a)** pre-failure, **(b)** bedrock, **(c)** post-failure and **(d)** up-to-date. The spatial resolution of the DEMs is 30 m in **(a–c)** and 1 m in **(d)**.

## 4.3 Distinct element modelling results

### 4.3.1 Modelling scenario I: initial deformation and failure mechanism of the rockslide

**Model geometry, boundary conditions and material properties**

As a basis, Profile 2 (see Figs. 2 and 3) was chosen to study the initial deformation and failure mechanism based on the Universal Distinct Element Code (UDEC; Itasca, 2020) by modelling both deformation and failure of blocks, as well as shearing and opening of joints. As input the reconstructed pre-failure topography was taken to create the surface of the slope. Since this model type focuses on the initial formation mechanism of the rockslide, the field-based and re-

constructed basal shear zone was not included. However, the mapped structural anisotropy was considered by building a fractured rock mass model based on vertical and eastwards dipping joints (dip angle of 32°). Both joint sets are fully persistent and are spaced at 50 m. A finite-difference mesh was calculated for deformable blocks by a zone edge length of 10 m and a rounding length of 0.3 m. This avoids the problem of contact overlap possibly resulting from the interaction of blocks occurring close to or at two opposing block corners (UDEC; Itasca, 2020). A Mohr–Coulomb constitutive model was chosen from UDEC's plastic model group to simulate block deformation and failure. The failure envelope for this model corresponds to the Mohr–Coulomb criterion with a tension cut-off (tensile yield function). The shear flow rule is non-associated, and the tensile flow rule is associated (UDEC; Itasca, 2020). Input requirements comprise as pa-

rameters the elastic bulk and shear modulus, the rock density, the cohesion and internal angle of friction, and in some cases the tension limit. For joints the Coulomb slip area contact model was taken to calculate shear displacement and slip. All selected physical, mechanical and hydraulic properties are summarized in Table 1.

No displacement boundaries were applied on the left, right and lower model boundaries (Fig. 8a). The in situ stresses were initialized in terms of a vertical gradient based on gravity and the horizontal stresses being half of the vertical stresses by assuming a $k$-ratio of 0.5. At the model origin (0, 0) the two horizontal stresses were assigned to $\sigma_{xx} = \sigma_{zz} = 25.6$ MPa and the vertical stress to $\sigma_{yy} = 51.2$ MPa. In models with groundwater flow a groundwater table was assumed with respect to characteristic groundwater flow patterns where the unsaturated zone between the surface and the water table is typically deep at the head of the slope, whereas the water table at the base of the slope is close to or at the surface (Fig. 8a, e.g. Fetter, 2001). For models calculating groundwater flow, the lower model boundary was set to no-flow (impermeable boundary). The left and right groundwater model boundaries were defined by a hydraulic gradient based on hydrostatic water pressure according to the assumed water table. The maximum water pressure was set to 19.72 MPa at the left boundary along the $y$ axis and to 8.96 MPa at the right boundary along the $y$ axis (Fig. 8a).

In order study the deformation and failure characteristics of the rock slope, selected block and joint parameters were varied. Concerning the elasto-plastic blocks, cohesion was set to 0.1 and 1 MPa, internal friction angle to 20, 25, 30, 35 and 40°, and tensile strength to 0 and 0.1 MPa. Joint cohesion and tensile strength was set to 0 MPa by varying the friction angle between 25, 30 and 35° (Table 1). This model scenario focuses on intact block failure, its location and spatial arrangement, and the type of failure (i.e. tensile or shear failure) and provides insights into the general mechanisms of slope failure and formation of the rockslide geometry, as well as into the initiation and progressive formation of a continuous basal shear zone.

### Model scenario I – without groundwater flow

For this type of model run a comprehensive parameter study was performed by varying block plasticity and joint properties (see above and Table 1). In Fig. 8b the spatial distribution of block displacement is shown, indicating a continuous decrease in magnitude from surface to depth primarily caused by block deformation. It is also evident that localized line-shaped zones (e.g. several internal shear zones) were formed, which suggests progressive fracturing and loosening of the rock mass most likely penetrating from shallow to deeper domains (Fig. 8d). A comparison with the location of the reconstructed basal shear zone indicates that simulated slope deformations are not penetrating to depths which are deep enough to reproduce the slope situation (i.e. location of

the basal shear zone). However, the shape of the rockslide was adequately reproduced. By analysing the shear displacement of joints and shear failure pattern of blocks (Fig. 8d) in the rock mass, it was determined that a combination of structurally driven shear displacement and block failure was responsible for the observed slope deformation characteristics. Increased magnitudes of shear displacement were observed at the inclined joints in the middle and lower parts of the slope, reaching depths almost down to the location of the basal shear zone (Fig. 8c). In addition, large shear displacements were also observed near the summit on vertical joints presumably induced by extensional stress regimes and rock mass subsidence. The pattern of block shear failure zones presented in Fig. 8d clearly indicates the formation of several persistent shear zones, with failure processes occurring particularly frequently at the foot of the slope and in the summit area. Again, the deepest shear zone formed is too shallow and does not reach the location of the real shear zone except at the foot of the slope. Results presented in Fig. 8b, c and d are based on the input parameters for blocks of TS3 $c_b = 0.1$ MPa and $\varphi_b = 30°$ and for joints of $c_j = 0.1$ MPa and $\varphi_j = 25°$. Changing the friction angle of the joints to TS4 25 or 35° while keeping the other parameters constant has no major impact on the modelling results. However, when varying the block friction angle to input values of 20, 25, 30, 35 and 40°, in which the block cohesion remains constant at $c_b = 0.1$ MPa, the results obtained are different. Whereas a block friction angle between 20 and 35° led to ongoing rock mass displacements reaching tens of metres, model runs with $\varphi_b = 40°$ stabilize after a few metres. Furthermore, the sensitivity of the model behaviour to varying block cohesion between 0.1 and 1 MPa is expressed in strongly different displacement magnitudes, i.e. 50 versus 1.7 m, and is characterized by shallow block displacements (Fig. 8e, f). In contrast, we have identified only a minor influence of block tensile strength on model behaviour. This is evident because tensile failure occurred primarily near the surface and at shallow depths (Fig. 8d).

### Model scenario I – with groundwater flow

Model runs considering groundwater flow show similar results as dry models (Fig. 8). It has to be mentioned that due to UDEC's limitations, water pressure is only applied to joints but not to blocks. A block friction angle of $\varphi_b = 30°$, while $c_b = 0.1$ MPa and the shear properties of the joints were set to $c_j = 0$ MPa and $\varphi_j = 30°$, caused a similar spatial distribution of block shear failure zones as observed under no groundwater flow conditions (compare Fig. 8d and g). Multiple shear zones at different depths were also created in this model, an indication that intensive fracturing and loosening processes of the rockslide mass occurred. However, the simulation results did not confirm the hypothesis of a single basal shear zone located at the trace of the reconstructed basal shear zone. Remarkably, the model run shown

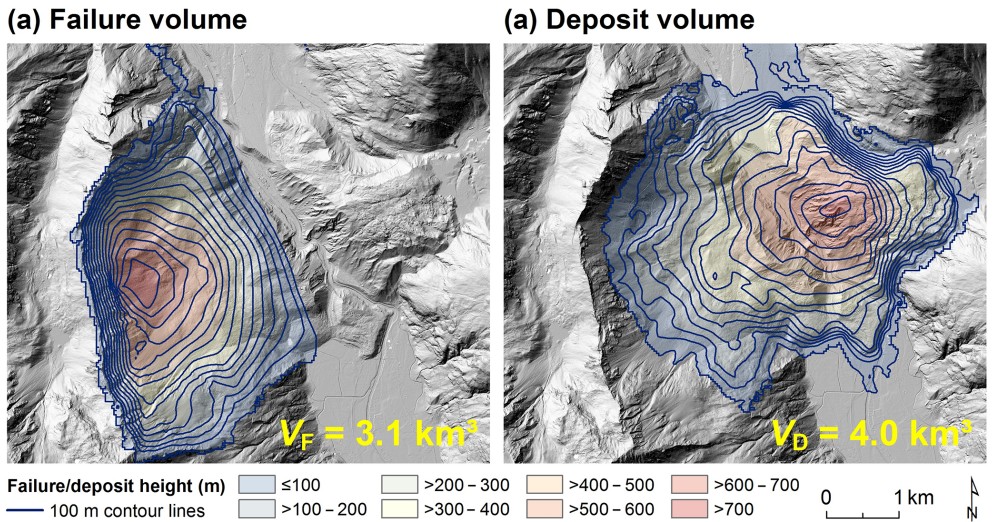

**Figure 7. (a)** Failure and **(b)** deposition heights and volumes of the Köfels rockslide mass computed in ArcGIS. The contour lines indicate the height difference between the **(a)** pre-failure and **(b)** post-failure topography and the topography of the sliding surface.

**Table 1.** Rock block and discontinuity properties for the distinct element modelling study for scenarios I and II, categorized into rockslide mass, underlying rock mass, basal shear zone and fractured rock mass. Hydraulic parameters ($a_{zero}$, $a_{res}$ and $j_{perm}$) are only relevant for model runs considering groundwater flow. Modelling scenario II is based solely on an elastic constitutive relationship for blocks, whereas for scenario I the blocks are simulated by applying a Mohr–Coulomb constitutive model.

| Model scenario | Scenario I | Scenario II | | |
|---|---|---|---|---|
| Material property | Fractured rock mass | Rockslide mass | Underlying rock mass | Basal shear zone |
| Block density, $\rho$ (kg/m$^3$) | 2600 | 2600 | 2600 | – |
| Block bulk modulus, $K$ (GPa) | 22 | 22 | 22 | – |
| Block shear modulus, $G$ (GPa) | 17 | 17 | 17 | – |
| Block cohesion, $c_b$ (MPa) | 0.1, 1 | – | – | – |
| Block internal friction angle, $\varphi_b$ (°) | 20, 25, 30, 35, 40 | – | – | – |
| Block tensile strength, $\sigma_t$ (MPa) | 0, 0.1 | – | – | – |
| Discontinuity normal stiffness, $j_{kn}$ (GPa/m) | 100 | 100 | 100 | 100 |
| Discontinuity shear stiffness, $j_{ks}$ (GPa/m) | 100 | 100 | 100 | 100 |
| Discontinuity cohesion, $c$ (MPa) | 0 | 0 | 0 | 0 |
| Discontinuity friction, $\phi_j$ or $\phi_{bs}$ (°) | 25, 30, 35 | 20, 30, 40 | 40 | 20, 21, 22, 23, 24, 25, 26, 27, 28, 29, 30, 31 |
| Hydraulic aperture for zero normal stress, $a_{zero}$ (m) | 0.00026 | 0.00018 | 0.00026 | 0.00026 |
| Hydraulic residual aperture, $a_{res}$ (m) | 0.00026 | 0.00018 | 0.00026 | 0.00026 |
| Discontinuity permeability constant, $j_{perm}$ (1/Pa s) | 83.3 | 83.3 | 83.3 | 83.3 |

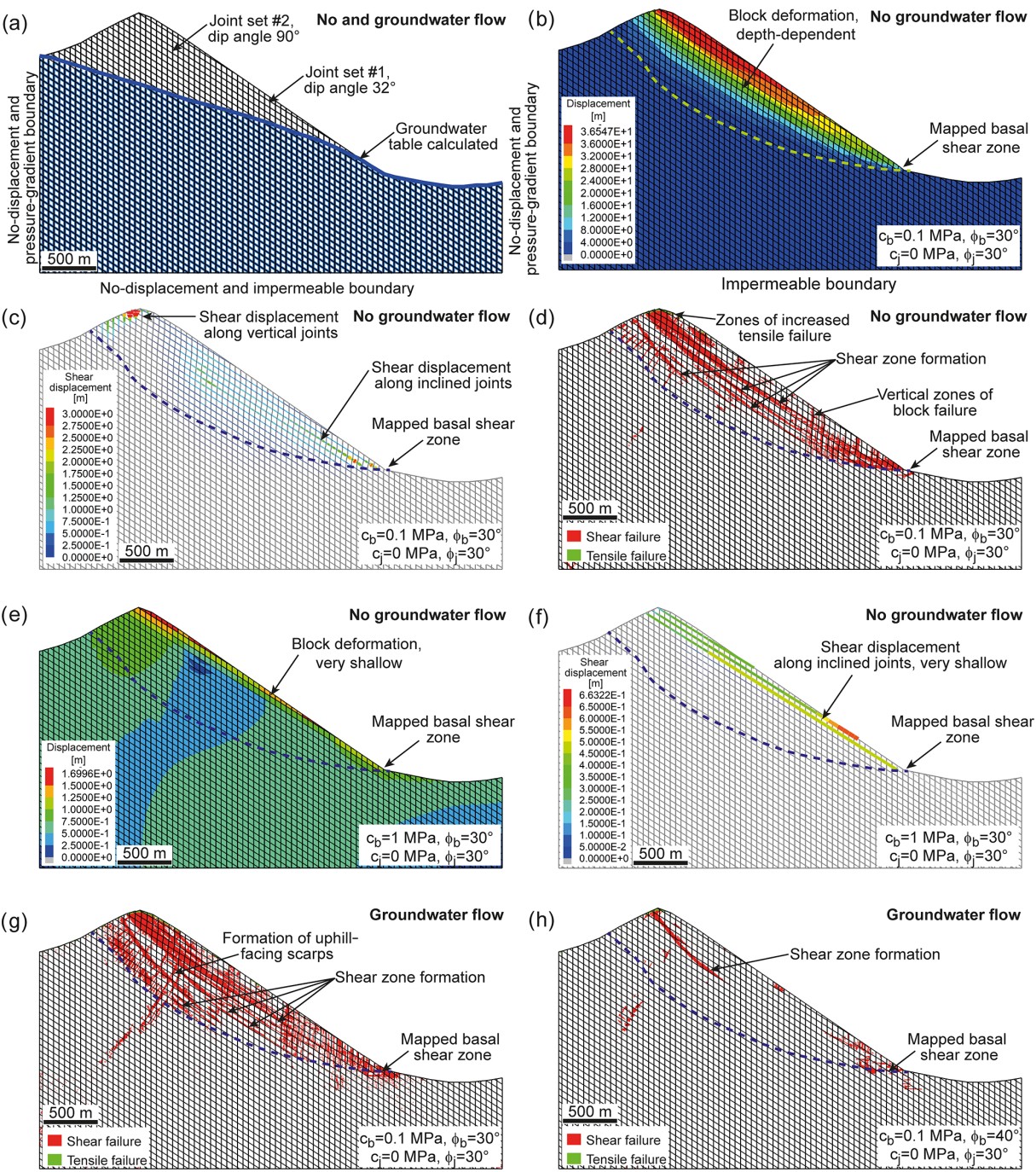

**Figure 8.** TS1 Results of modelling scenario I: **(a)** model set-up presenting the joint network and the calculated groundwater situation, **(b)** depth-dependent distribution of block displacements by implementing a block cohesion of $c_b = 0.1$ MPa and internal friction angle of $\phi_b = 30°$, **(c)** spatial distribution of shear displacements along joints ($c_b = 0.1$ MPa, $\phi_b = 30°$), **(d)** formation of multiple shear and antithetic zones due to block shear failure in a line-shaped arrangement ($c_b = 0.1$ MPa, $\phi_b = 30°$), **(e)** model run showing very shallow block displacements when block cohesion was increased from $c_b = 0.1$ to $c_b = 1$ MPa ($\phi_b = 30°$), **(f)** shallow-occurring shear displacements along joints for $c_b = 0.1$ MPa TS2 and $\phi_b = 30°$, **(g)** model run considering groundwater flow showing the formation of multiple shear and antithetic zones (i.e. uphill-facing scarps) due to block shear failure ($c_b = 0.1$ MPa, $\phi_b = 30°$), and **(h)** model run with groundwater flow showing the formation of a single shear zone being initiated mainly in the summit area and at the foot of the slope when the friction angle is increased to $\phi_b = 40°$ ($c_b = 0.1$ MPa). All model runs presented reached stabilization.

in Fig. 8g clearly indicates the formation of antithetic shear zones, i.e. structures which were often observed in the context of deep-seated rockslides and are appearing on the surface as uphill-facing scarps. Whereas block displacements of the previous model reached magnitudes of several decametres, increasing the block friction angle to $\varphi_b = 40°$ while leaving the other parameters unchanged reduced the overall displacement to less than 3.5 m. Even though slope displacements were rather small, the simulation result suggests the initiation of a shear zone mainly developing near the summit and less apparent near the foot (Fig. 8h). Nevertheless, the depth of the shear zone is clearly too small to be consistent with on-site observations.

### 4.3.2 Modelling scenario II: back-calculation of shear strength properties of the basal shear zone

**Model geometry, boundary conditions and material properties**

Profile 2 (see Figs. 2 and 3) was also taken for the back-calculation study based on distinct element modelling (UDEC; Itasca, 2020) with a fully persistent basal failure zone. The pre-failure topography was implemented to create the surface of the slope, whilst the topography of the sliding surface provides the input for the basal shear zone (Fig. 9). The main deformation within the system takes place through the movement along the basal failure zone (i.e. sliding and rotation of blocks, as well as opening and interlocking of interfaces). In order to obtain numerical models that are manageable from the point of view of computer performance and computation time, the spacing of the joint sets was specified to 50 m in the rockslide mass and 150 m in the underlying bedrock (Fig. 9a and c). Two types of discontinuity networks were studied. The first type is characterized by a fully persistent vertical and horizontal joint set and the second type by a fully persistent vertical and inclined joint set (dip angle of 32°). One reason to use a discontinuity geometry based on a vertical and horizontal fully persistent joint set was to have good control over the groundwater flow conditions in the models by achieving isotropic hydraulic conductivity. However, a structurally more realistic model geometry was added to this study. The finite-difference mesh in the model was assigned a size of 20 m in the rockslide mass and 50 m in the underlying granitic gneiss block. Roundings of block corners were applied with a radius of 0.3 m. The mechanical and hydraulic boundaries, as well as the groundwater table, were defined the same as in modelling scenario I (Fig. 9b, d). Blocks were considered as linear elastic as defined by Hooke's law, considering components of stress to be linear functions of components of strain (Jaeger et al., 2007). Physical, mechanical and hydraulic properties used for the simulation are shown in Table 1. The Coulomb slip area contact model was assigned to the two joint sets and the basal shear zone. To investigate the impact of the discontinu-

ity network on the back-calculated friction angle of the basal shear zone, the shear strength and tensile strength properties of the joints were varied in the rockslide mass (Table 1). The friction angle of the basal shear zone was varied between 20 and 28° for models without groundwater flow and between 25 and 31° considering groundwater flow. As an assumption, the cohesion of the basal shear zone was set to $c = 0$ MPa for all models. The determination of the critical angle of friction of the basal shear zone at failure was done by monitoring the maximum shear displacement along the basal shear zone, the block displacements (see monitoring points in Fig. 9a and c) and the unbalanced forces of the model, ideally reaching values close to zero.

**Model scenario II – without groundwater flow**

Concerning the model type without groundwater flow, the friction angle for the basal shear zone was varied between 20 and 38° TS5, whilst all other parameters were kept constant. However, to study also the impact of internal rockslide deformability, the friction angle of the discontinuity network was set to 20, 30 and 40°. As a result, the back-calculated critical friction angles where failure was beginning were not a single value but rather a range varying from 21 to 24°. In addition, it was found that a stepwise reduction of the friction angle led to increasing displacements, reaching nearly 6 m in the transition zone, associated with a re-stabilization of the rockslide mass (Fig. 10a, b, c and d). Only a further reduction of the friction angle finally led to the progressing rockslide displacement without stabilization. One major factor influencing the back-calculated critical friction angle was related to the shear strength properties of the joints in the rockslide mass affecting the deformability (Fig. 11). On the one hand, a joint friction angle of $\varphi_j = 40°$ increases overall stability, which in turn requires a low friction angle of the basal shear zone values of $\varphi_{bs} = 21°$ to simulate failure. On the other hand, a rather low value of $\varphi_j = 20°$ assigned to the joints of the rockslide mass increase the critical friction angle of the basal shear zone to $\varphi_{bs} = 24°$. A value in between, i.e. $\varphi_j = 30°$, results in a critical value of $\varphi_{bs} = 23°$ for the basal shear zone. Further, it was found that the influence of the discontinuity orientation on slope deformation behaviour is rather small. Based on a joint friction angle of $\varphi_j = 40°$, no difference in the displacement behaviour was observed between the model type with vertical and horizontal joints and the model type with vertical and inclined joints.

The magnitude and spatial distribution of shear displacement is affected by the shape of the basal shear zone (Fig. 10b, d). Exemplarily, for the model types shown in Fig. 10b and d ($\varphi_j = 30°$, $\varphi_{bs} = 22°$) the largest shear displacements were obtained in the upper part of the slope along the steepest section of the basal shear zone, reaching about 5.8 and 5.7 m. Towards the upper and lower sections of the shear zone, shear displacement continuously reduces to values of 3.75 m. The rockslide mass located above the steeply

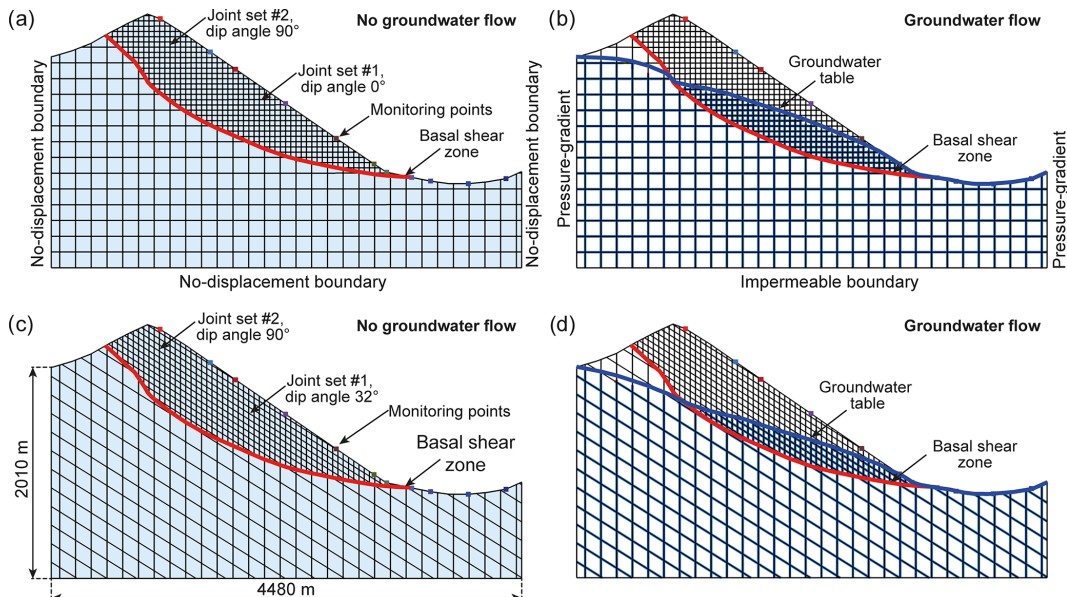

**Figure 9.** Model set-up and groundwater conditions of modelling scenario II: **(a)** model geometry characterized by a fully persistent horizontal (dip angle 90°) and vertical (dip angle 0°) joint set and the reconstructed basal shear zone (no groundwater flow), **(b)** same model geometry as in **(a)** but by considering groundwater flow, **(c)** model geometry characterized by a fully persistent inclined (dip angle 32°) and vertical joint (dip angle 0°) set and the reconstructed basal shear zone (no groundwater flow), and **(d)** same model geometry as in **(c)** but by considering groundwater flow. Monitoring points are included to prove if model runs have reached equilibrium (unbalanced forces) and stabilization (i.e. no ongoing displacements).

inclined section represents the domain which is most affected by internal deformation, i.e. due to joint shearing ($<2$ m) and opening ($<2$ m), thus clearly indicating the largest block displacements (Fig. 10a and c). The shear to normal stress ratio ($\tau_s/\sigma_n$) along the basal shear zone fluctuates slightly over the entire length, and the calculated $\tau_s/\sigma_n$ value is the range of the coefficient of friction, i.e. $\mu_{bs} = \tan(\varphi_{bs})$, applied to the model.

## Model scenario II – with groundwater flow

The effect of groundwater flow due to water pressure in the discontinuity network and basal shear zone on the back-calculated shear strength parameters was investigated for the two model types characterized by horizontally or inclined joint sets. In Fig. 9b and d it is noticeable that the groundwater table has a kink with a steeper hydraulic gradient at the transition from the undeformed bedrock to the rockslide mass. In addition, results showed a difference in the joint water pressure distribution between the two geometric model types (Fig. 9b and d). Although the same spacing and hydraulic aperture values were used, the model type with the inclined joint set resulted in a lower water table accompanied by reduced water pressures. The reason for this can be found in the structural anisotropy changing the hydraulic conductivity from isotropic to anisotropic conditions.

Similar to the model runs without groundwater flow (i.e. horizontal and vertical joint sets), the variation of the joint

friction angle of the rockslide mass to values of 20, 30 and 40° influenced the back-calculated friction angle of the basal shear zone (Fig. 11). A joint friction angle of $\varphi_j = 40°$ results in a critical basal shear zone friction angle of $\varphi_{bs} = 27°$. The reduction of the joint friction angle to $\varphi_j = 30°$ and $\varphi_j = 20°$ requires higher critical basal shear zone values of $\varphi_{bs} = 28°$ and $\varphi_{bs} = 30°$, respectively, to simulate failure. Thus, depending on the possibility of shearing along joints, the back-calculated friction angle of the basal shear zone varies between 28 and 31°. Interestingly, as opposed to the friction angle of the joints, the impact of tensile strength on the critical friction angle of the basal shear zone was minor. For example, changing the tensile strength to $\sigma_z = 1$ MPa while maintaining the joint friction angle at $\varphi_j = 40°$ showed nearly no effect on the back-calculated critical friction angle of the basal shear zone.

One major difference in comparison to the no groundwater flow models is related to the spatial distribution of shear displacement along the basal shear zone, as well as overall rockslide displacement (Fig. 10e and f). For model runs with horizontal and vertical joint sets, shear displacement continuously increases from the foot of the slope to the scarp area, reaching the largest values at the steepest section or in some cases above it (Fig. 12). In contrast, the models assigned an inclined joint set indicate an opposing trend, with the largest shear displacements near the foot of the slope (Fig. 12), suggesting that geological structures influence the spatial distribution of shear displacement along the basal shear zone.

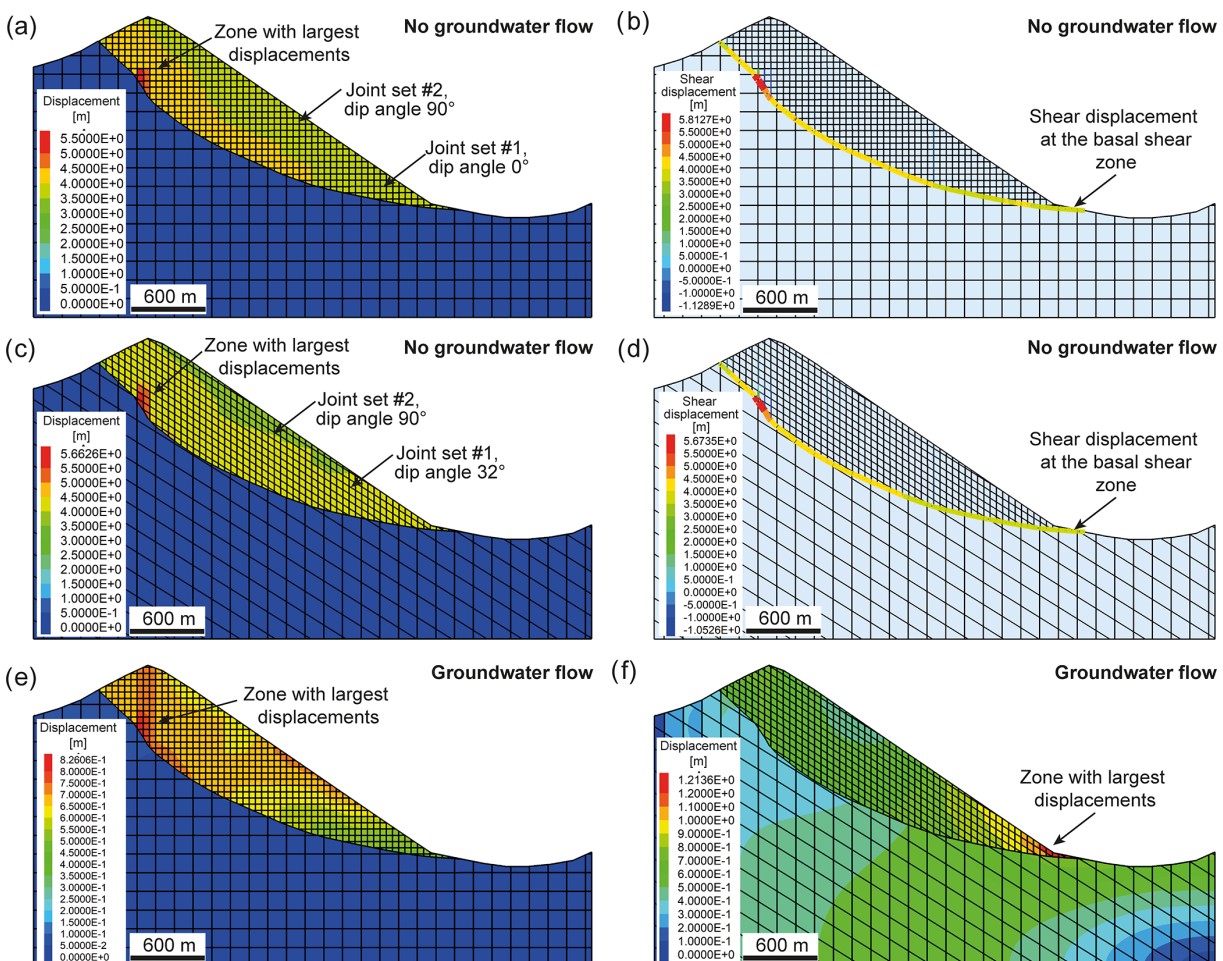

**Figure 10.** Results of modelling scenario II: **(a)** internal deformation of the rockslide mass with horizontal and vertical joint sets presented by the maximum block displacements around the steeply inclined section of the basal shear zone, **(b)** shear displacements along the basal shear zone for the same model type shown in TS6 panel **(a)**, **(c)** internal deformation of the rockslide mass with horizontal and inclined joint sets presented by the maximum block displacements around the steeply inclined section of the basal shear zone, **(d)** shear displacements along the basal shear zone for the same model type as in **(c)**, and **(e)** internal deformation of the rockslide mass with horizontal and vertical joint sets when groundwater flow is considered. Similar to dry conditions, maximum block displacements were calculated near the steeply inclined section of the basal shear zone, and **(f)** internal deformation of the rockslide mass with horizontal and inclined joint sets when groundwater flow is considered. Maximum block displacements were calculated near the foot of the slope.

Concerning the spatial distribution of displacements of the rockslide mass, a similar behaviour was obtained from the simulations. Whereas the model runs with horizontal and vertical joints yielded the largest displacements in the upper part of the rockslide, the opposite trend, with the maximum displacements at the foot of the slide, was observed for the models with inclined joints (Fig. 10e and f).

In summary, for model scenario II the critical friction angles are ranging from 21 to 24° when there is no groundwater flow and from 27 to 30° when groundwater flow was considered.

## 5   Discussion

### 5.1   Reconstruction of rockslide topography, geometry, volume and sliding mass porosity

Topographic reconstructions and volumetric and porosity calculations of the failure and deposition mass of the Köfels rockslide have been made before by Brückl et al. (2001). They determined physical properties of the rock mass from seismic data. Based on an empirical relationship given by Watkins et al. (1972) the P-wave velocities were plotted versus depth to estimate the porosity of the deposition mass. On the basis of this calculation a relation between the thickness of the overburden and the porosity of the deposits was developed. The calculations by Brückl et al. (2001) resulted

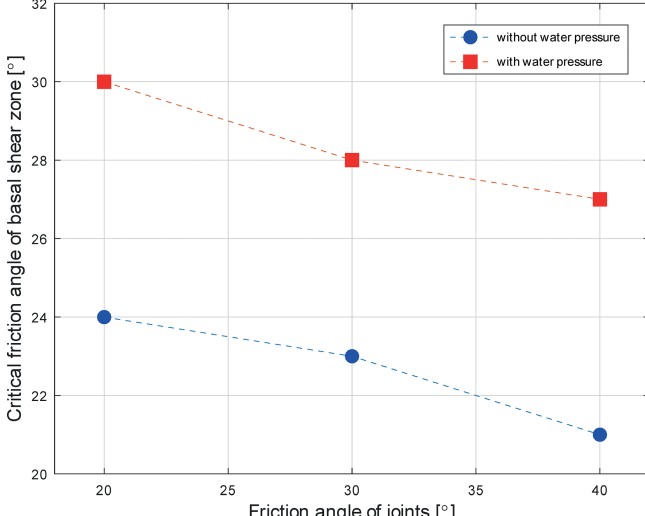

**Figure 11.** Relationship between the applied friction angle of the joints in the rockslide mass and the back-calculated critical friction angle of the basal shear zone at failure, grouped into model runs without and with groundwater flow (cohesion was set to zero for joints and the basal shear zone).

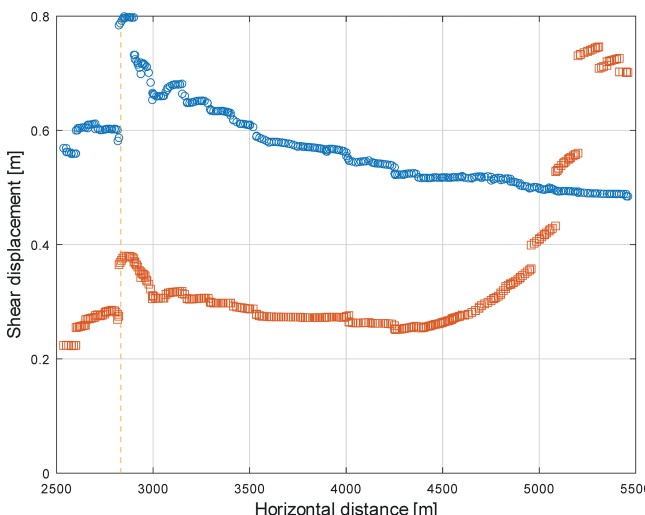

**Figure 12.** Spatial distribution of shear displacement along the basal shear zone for the model types with horizontal and vertical joint sets (blue circles; see Fig. 10e) and inclined (32°) and vertical joint sets (red squares; see Fig. 10f). At model location $x = 2823$ m, a step in shear displacement occurred due to a kink in the basal shear zone. Both model types are based on groundwater flow (see Fig. 9b, d).

in a post-failure mean depth independent porosity of 23 %. With an estimated failure volume of 3280 million m³ and a deposition volume of 3880 million m³, Brückl et al. (2001) calculated a volume increase of 18 % due to disintegration, fracturing and loosening of the rock mass by the sliding process. The volume increase obtained by this study is 29 % and therefore remarkably larger than that obtained by Brückl et

al. (2001). This discrepancy indicates that the computed volume increase is very sensitive to the computed failure and deposition volumes. Concerning the porosity of the deposition mass we calculated a value of 26 %, a value similar to the 23 % of Brückl et al. (2001).

## 5.2 Geomechanical modelling

In this study a topographic reconstruction was performed to provide a reasonable pre-failure, post-failure and geometrical model of the Köfels rockslide for subsequent numerical modelling. Given that slope inclination and rockslide geometry have a large impact on stability and on limit equilibrium, the detailed reconstruction of the pre-failure and post-failure slopes and the slide geometry done herein made possible comprehensive cross-checks and plausibility checks.

Distinct element modelling of the fractured rock slope without implementing a basal shear provides insight into the initial deformation and failure processes. Multiple shear zones at different depths were created in these models, suggesting intensive fracturing and loosening of the rockslide mass during the initial phase of evolution rather than forming a single basal shear zone. In addition, the formation of antithetic shear zones was observed during several model runs. Both features are typical for deep-seated rockslides and were frequently mapped on surface as either downhill- or uphill-facing scarps or explored in the subsurface by boreholes as shear zones composed of low-strength fault breccias and gouges (Rechberger et al., 2021; Strauhal et al., 2017). The formation of slabs is a further indicator for increased fracturing and loosening of the rockslide mass, and it was determined by several case studies in crystalline rocks (Bonzanigo et al., 2007; Glueer et al., 2019; Zangerl et al., 2019). The obtained rock mass strength degradation process can be understood as the consequence of a complex interaction between pre-existing joints and brittle fracture propagation through intact rock bridges, herein modelled as block failure. This modelling campaign does not take into account time-dependent processes and intact rock failure and crack propagation based on fracture mechanics. However, the simulation of zones of increased shear and tensile failure in blocks on the basis of a Mohr–Coulomb constitutive model takes into account the concept of progressive failure, at least to some extent. According to Eberhardt et al. (2004) progressive failure in fractured rock slopes is related to the failure of individual rock bridges as their shear strength is exceeded. This in turn would increase the stresses ahead of the shear plane causing subsequent intact rock bridge failure in a consecutive manner until the rupture surface extends to the point where kinematic release is possible. Although progressive failure must have played a crucial role in the genesis of the Köfels rockslide, additional factors must still have been involved to lead to the failure of such a strong rock mass.

The complex interaction of deformation and failure processes of blocks and joints was investigated by our com-

prehensive parameter study based on more than 50 different model runs. Large slope displacements associated with the formation of shear zones (i.e. block failures) were induced when the cohesion of the blocks is reduced to $c_b = 0.1\,\mathrm{MPa}$ and the internal friction angle to $\varphi_b = 35°$ by applying a friction angle of $\varphi_j = 30°$ to the joint network. Increasing the internal friction angle of the blocks to $c_b = 1\,\mathrm{MPa}$ requires in turn a significant reduction in the friction angle to $\varphi_b = 20°$ in order to simulate slope failure. In comparison, empirical estimations of the rock mass strength based on field surveys and the application of the GSI approach by Hoek and Brown (1997) show that the obtained shear strength values of the rock mass of $c_{rm} = 2\,\mathrm{MPa}$ and $\varphi_{rm} = 35°$ are too high to promote slope failure under static conditions.

The back-calculation of shear strength of the basal shear zone based on UDEC (Itasca, 2020), assuming a cohesion of zero, in the present study results in values of $\varphi_{bs} \leq 21$–$24°$ without pore water pressure and $\varphi_{bs} \leq 27$–$30°$ with pore water pressure. Consequently, given that the imposed boundary conditions, rock mass parameters and water pressure are valid, the friction angle of the basal shear zone may be constrained to the range of $\varphi_{bs} = 21$–$30°$. This very wide range is a result, on the one hand, of the influence of water pressure and, on the other hand, of the internal deformability of the rockslide mass, primarily controlled by shearing along joints. Brückl and Parotidis (2001) obtained a value of the rock mass friction angle ($\varphi_{rm}$) between 20 and 24° from geomechanical continuum modelling of the Köfels rockslide. In a later approach, Brückl and Parotidis (2005) modelled the Köfels rockslide by applying a 2D finite-element method and by focussing on the modelling of the rockslide failure geometry. For their approach, they assumed a friction angle of $\varphi_{bs} = 28°$ without considering pore water pressure.

A preliminary comparison with 3D limit-equilibrium slope stability models (r.slope.stability; Mergili et al., 2014a, b) indicates that the critical safety factors yielded by these models are within the range gained by the discontinuum approach, suggesting a certain degree of plausibility of the simulations. However, more research is necessary to explore this issue and particularly the influence of using 3D models.

Concerning unfilled and rough joints in granitic rocks the friction angle guessed by Barton and Choubey (1977) is several degrees higher than that needed for failure. In situ shear strength data for rough, unfilled joints in granitic rocks published by Fishmann (2004) show a remarkably high friction angle of $\varphi_j = 44°$ linked to a cohesion of $c_j = 0.08$ and $c_j = 0.14\,\mathrm{MPa}$. Grøneng et al. (2009) determined the shear strength of unfilled rock joints focusing on the Åknes rockslide in Norway by applying the Barton–Bandis empirical equation. Applying their proposed parameters to an in situ stress range from 4 to 18 MPa results in a friction angle between 31 and 36°. Grasselli (2001) performed shear tests on fresh tensile rock joints of gneisses and granites, additionally by applying up to six shear test cycles on the same samples. He measured values between 39 and 69° for the peak friction

angle and values between 35 and 57° for the residual friction angle. Both Byerlee (1978) and Hencher et al. (2011) determined a (basic) friction angle of around 40° for granitic rock joints.

In addition, field observations confirm that the fractures have a persistence on the scale of metres to tens of metres, and therefore it is assumed that the rock mass at the Köfels rockslide is strengthened by intact rock bridges. It is widely accepted that intact rock bridges, if present, increase the shear strength of a rock mass (Jennings, 1970; Einstein et al., 1983). Intact rock bridge failure is complex, is usually not simply related to in-plane shear along the fractures and is characterized by time-dependent progressive failure processes (sub-critical crack growth; Atkinson 1984, 1987). Nevertheless, conceptually and in the context of the Köfels rockslide intact rock bridges would further increase the overall rock mass strength of the slope also when considering long-term conditions and the concept of progressive failure. So far, the only possible geological discontinuity type which displays shear strength properties low enough to allow for slope failure under static conditions are pre-existing brittle fault zones composed of infillings of gouge and breccia. The parameters back-calculated by numerical modelling correspond reasonably well with the bandwidth of published values, ranging from 19 to 30°, observed for shear zones in crystalline rocks (Engl et al., 2008; Strauhal et al., 2017). However, our detailed geological field investigation and structural analyses of the high-resolution digital elevation models could indeed identify such structures in the rock mass but not those ones which are dipping moderately towards the east and are thus favourably aligned to promote the rockslide formation.

For example, an additional geological factor reducing the rock mass strength by rock mass fracturing and weakening of large rock slopes to depths of several hundred metres is related to deep-seated block or flexural toppling processes (Amann, 2006; Casson et al., 2003; Zangerl et al., 2015). Deep-seated toppling occurs when steeply inclined structures are present, and this failure mechanism is often observed in a foliated metamorphic rock mass of low to moderate strength (paragneisses, schists and phyllites). However, less common but still observed is toppling in granitic gneisses when foliation, joint planes and fault zones are closely spaced and steeply dipping into the slope (Amann, 2006). Structural mapping by Prager et al. (2009) and this study in the surroundings of the scarp confirm steeply dipping northnorthwest–south-southeast striking joints and faults. Though structurally possible, it is questionable if deep-seated toppling is a preparatory mechanism for the Köfels rock mass failure because no clear geomorphological and structural indicators for toppling were found in the surroundings of the scarp.

Based on the results of the numerical modelling study it is inconceivable that slope failure occurred under pure static conditions even when a high groundwater table causing extraordinary high pore pressures is assumed. Apart from the

fact that permafrost degradation due to climate warming has often been discussed as a relevant factor for slope failure in rock masses, so far it is widely accepted that permafrost degradation can alter the rock mass strength by ice melting and temperature changes (Dramis et al., 1995; Fischer et al., 2006; Huggel et al., 2012; Krautblatter et al., 2013). According to the time-depended rock-ice mechanical model proposed by Krautblatter et al. (2013), it is assumed that ice-rock mechanical processes are more relevant for rock slope failures at shallow depths (less than 20 m), whereas rock-rock mechanical processes are dominating at greater depths. Considering this geomechanical concept in relationship to the great depth of the rupture surface of several hundred metres, as well as the time lag of two millennia between Holocene warming and slope failure (Nicolussi et al., 2015), permafrost degradation acting as the major trigger of the Köfels rockslide is unlikely. Based on the findings of this study, climate-driven triggering factors characterized by periods of increased precipitation rates or permafrost degradation were probably too weak to provoke such a large-scale slope failure.

One interesting observation was made by Nicolussi et al. (2015) who performed precise age dating of the 3100 million m$^3$ large Köfels rockslide based on tree-ring analysis and radiocarbon dating, constraining the event to 9527–9498 cal BP. Remarkably, the new age bandwidth is close to the age of the Flims landslide ranging from 9480–9430 cal BP, the largest rockslide in the Alps, comprising a volume of 8000–12 000 million m$^3$ (Poschinger and Kippel, 2009). Furthermore, a few more events occurring in the eastern Alps show ages clustering within this period (Prager et al., 2008; Borgatti and Soldati, 2010). The close temporal and spatial relationship between the Köfels and Flims rockslide raises the question of whether dynamic loading due to earthquake shaking was able to trigger two of the largest rockslides in the Alps, located only about 130 km apart. Oswald et al. (2021) concluded from high-resolution lacustrine palaeoseismology a relation between past seismicity and a spatiotemporal cluster of large prehistoric rockslides in the eastern Alps, e.g. the Eibsee, Fernpass and Tschirgant rockslides (Prager et al., 2008). They also found that the Köfels rockslide was not directly earthquake-triggered but failed some centuries later after at least one severe earthquake around 9.9 ka BP. Kremer et al. (2020) proposed enhanced seismic activity in the Alps in the period of 9500–9900 cal BP. Based on the chronology of earthquake events during the Holocene, Oswald et al. (2021) assume that earthquakes are more important for preparing rock slopes for failure due to seismic fatigue than for being the ultimate trigger. In the context of this numerical modelling study progressive failure reinforced by seismic fatigue can explain to some extent the discrepancy between the rock mass strength estimated from the rock mechanical assessment and that obtained by back-calculation. However, seismic fatigue cannot solely explain the particular situation of the Köfels rockslide because it is still unclear why this giant event occurred at this location and within a very strong rock mass. With regard to the surrounding area, the reconstructed pre-failure slope is neither particularly steep nor characterized by other eye-catching features. However, it is assumed that besides the other factors a special geological predisposition (e.g. favourably orientated and very high persistent discontinuities) may have contributed to the occurrence of the event.

## 6 Conclusions

Based on geologic, geophysical and topographic constraints, we reconstructed three topographic stages of the Köfels rockslide: (i) the pre-failure topography with the reconstructed mountain summit, (ii) the topography demonstrating the sliding surface without rockslide deposit and (iii) the post-failure topography with the deposits in the valley but before their incision by the river Ötztaler Ache. For the failure volume a value of 3100 million m$^3$ is obtained, the deposition volume is calculated at about 4000 million m$^3$. These values are very close to those derived by Brückl et al. (2001), leading to the conclusion that the estimates obtained of the volumes are sufficiently robust.

Knowledge on the volume increase in the rock mass during sliding is less robust as the derived values react very sensitively even to small variations in the failure and deposition volumes. Whilst Brückl et al. (2001) come to an increase in volume of 18 %, our study suggests an increase of 29 %. The porosity of the failed rockslide mass increased to a mean of 26 % with wide variations.

Based on distinct element models by varying the block and joint input parameters, the deformation and failure process of the rock slope could be plausibly reconstructed; however, the exact geometry of the rockslide, especially in view of thickness and location of the basal shear zone, could not be fully reproduced. Our results suggest that both failure of rock blocks and shearing along moderately eastward-dipping joints were responsible for the formation of the rockslide. The progressive failure process may have taken place by fracturing, fragmentation and loosening of the rock mass, advancing from shallow to deeper zones of the slope. Progressive rock mass degradation may have led to the formation of multiple shear zones at different depths and antithetic structures such as uphill facing scarps.

The shear strength of the basal shear zone at failure in conditions without and with pore water pressure is back-calculated by the distinct element method. The back-calculation study is based on the assumption of a continuous basal shear zone derived from field surveys and high-resolution digital elevation models and a cohesion of zero, resulting in values of $\varphi_{bs} \leq 21$–24° without pore water pressure and $\varphi_{bs} \leq 27$–30° with pore water pressure.

Field observations suggest that a continuous basal shear zone may have formed during the initial failure stage of the slide, but there is no evidence for a pre-existing zone of weakness promoting slope failure. Comparisons of back-calculated shear strength properties of the basal shear zone with values roughly assessed for the fractured granitic rock mass show that slope failure under static conditions is unlikely even under high pore pressures. Thus, a particular geological disposition and increased seismic activity over a longer period of time could be a major driver of the progressive strength degradation process responsible for the rapid failure leading to the Köfels rockslide. However, since DEM modelling can only consider some aspects of the expected rock mechanical processes, it makes sense to carry out additional numerical modelling studies based on other approaches and loading conditions. In addition, subsurface investigations and rock mechanics tests would help to better determine the parameters of the rock and the discontinuities, as well as the structure of the rock mass. Due to the exceptionally large volume of the rockslide, it should be examined if the classical laws of rock mechanics can fully represent the natural event.

*Code availability.* (1) Maps were created using ArcGIS® software version 10.2 by Esri. ArcGIS is the intellectual property of Esri and is used herein under license. Copyright © Esri. All rights reserved. For more information about © Esri software, please visit https://www.esri.com (Esri, 2014).

(2) Geomechanical numerical modelling was performed by UDEC – universal distinct element code. UDEC was used herein under license of ITASCA. For more information about UDEC, please visit https://www.itascacg.com/software/udec (Itasca, 2020).

(3) Stereonet pole and contour plots were made using the Stereonet software, version 10, by Rick Allmendinger, which is available at: http://www.geo.cornell.edu/geology/faculty/RWA/programs/stereonet.html (Allmendinger, 2018; Cardozo and Allmendinger, 2013), 2018.

*Data availability.* The study is largely based on the digital terrain model (DTM with a raster size of 1 m) provided by the Provincial Government of Tyrol and derivatives thereof. The authors are not entitled to make these data publicly available. Input and output files of the geomechanical modelling study can be obtained directly from the authors.

*Author contributions.* CZ contributed the general idea of the work and acquired the data, as well as the funding. AS and MM did the GIS-based analysis. GS performed the rock discontinuity mapping in the field and the discontinuity analysis. CZ planned and performed the geomechanical simulations based on UDEC. CZ, MM and AS wrote the manuscript, and all authors contributed to the illustrations and to the editing and revision process.

*Competing interests.* The authors declare that they have no conflict of interest.

*Acknowledgements.* This study was part of the alpS research projects "ProMM" and "AdaptInfra", which were supported by TIWAG, geo.zt, ILF Consulting Engineers and the Austrian Research Promotion Agency (COMET-program). The alpS-K1-Centre was supported by the federal ministries BMVIT and BMWFW, as well as the states of Tyrol and Vorarlberg in the framework of "COMET-Competence Centers for Excellent Technologies". COMET is processed through FFG.

*Review statement.* This paper was edited by Yves Bühler and reviewed by Gregor Ortner and two anonymous referees.

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

**Remarks from the language copy-editor**

CE1     Please note that it is our house standard to use numerals with all units, even when separated by another word. Without any change in meaning, this could be re-written "1 km more". Please advise.

**Remarks from the typesetter**

TS1     Why should this be changed? Values should not be adjusted at this stage without contacting the editor for approval. Also, with the change, the values in the figure would be different to those in the caption.

TS2     We have changed this back. We allowed it in the last round to match the figure but if you want to adjust the figure now too, the editor must approve the change first.

TS3     See previous comment on this issue.

TS4     See previous comment on this issue.

TS5     See previous comment on this issue.

TS6     We have added this word for more readability.

TS7     Please add last access date. I have added the link here.