# Peer review of "GIS-based topographic reconstruction and geomechanical modelling of the Köfels Rock Slide"

_Natural Hazards and Earth System Sciences, 2020_

## Short Comment (SC1) · 15 Sep 2020

Dear Sir, Can this research be used to predict future slippage in nearby sites, based on the results of this research? Kind Regards, Azealdeen Al-Jawadi

---

## Referee Comment (RC1) · Gregor Ortner (Referee) · 9 Oct 2020

All in all, it is a good and solidly constructed paper that contains a lot of work and effort. The authors had an unusually good initial situation regarding the available data and comparative studies. They have taken good advantage of it. The used approach is reasonably chosen and corresponds to the common standards and practice in this field and leads to an interesting outcome. And thus contributes to an increase of knowledge in the field of land slide analysis.

Specific comments

Basically the modeling strategy is clear and reasonable. To be able to reconstruct and model a mass movements in retrospect, it is clear that some assumptions must be

made. These assumptions have a major impact on the final result. As a reader I found it somewhat difficult to fully follow the choice of the assumptions and parameterization made for modelling. It would be desirable that the origin of a chosen parameters is clearly declared and referenced and the choice of the parameters is sufficiently justified.

This concerns especially the chapter "3.3.2 Model geometry, boundary and initial conditions" and "3.3.3 Material properties" as well as and the choice of water horizons in the model

- Model boundarys - Hydrostatic water pressure - Material properties

The choice of geotechnical parameters as well as the groundwater horizons should be addressed in the discussion and covered in a separate sub-section "Uncertainties" or "Model Uncertainties". In this subsection also the results of the modeling (with and without pore water pressure) should be discussed and critically reviewed.

Technical corrections

Line: 84: "exceptional high groundwater levels" -> It would be interesting to know how to reach this assumption Line:110 – 114: Mention when the landslide was Line:119: briefly mention where the value of the volume 3km3 originates from Line:127: "Tauferberg" in Fig.1 not not labeled in Fig.2 called "Tauferer Berg" please label uniformly Line:128: please label "Horlauchtal valley" in the figures Line:135-137: "This distinctive fragmentation of rock led to radon gas emissions and locally radioactive springs, which still affects today's population in Umhausen and causes noticably high cancer rates (Purtscheller et al. 1995)." -> interesting but irrelevant

Line:158 - 160: "To what extent permafrost degradation is able to trigger a deep-seated rock slides characterized by a shear zone at a depth of several hundred metres, is unclear and still under discussion (Nicolussi et al. 2015)." -> perhaps not even worth mentioning

Line:215: "Estimation of rock mass strength and shear strength of discontinuities was done" -> please briefly explain how

Figures: Style is all a bit "old school" but of course sufficient. Sometimes the labeling of Figs is not consistent with the text. Fig.1d: it would be nice if the jointsets are marked with colored lines and flags for the dip for better visualisation.
* * *

---

## Referee Comment (RC2) · Anonymous Referee #2 · 22 Oct 2020

General Comments

This manuscript deals with the preconditioning, preparatory and triggering factors of the largest catastrophic slope failure in crystalline rocks of the European Alps, called Köfels Rock Slide. The Köfels landslide shows abundant frictionites in the deposits, which makes this landslide a very interesting object for research carried out during the last 150 years. The manuscript focusses on frictional strength properties of the basal shear zone at the stage of complete slope failure. The investigation is based on a back analysis of static landslide stability, using the commercial distinct element code UDEC. The primary inputs to the model are the landslide geometry at the onset of failure, primarily derived from published data, and some assumptions about the elastic and strength properties of the intact rock and discontinuities. New data presented re-

fer mainly to the orientation of foliation and fractures mapped in the head scarp area. The results are well presented but are of limited significance, they mainly refer to the bulking factor and bulk friction angle along the assumed rupture plane, for dry and wet conditions. The reconstructed pre-failure topography and sliding plane location is similar to previous investigations. The estimation of the bulk friction angle is based on a set of assumptions which are critically reviewed in the next paragraph. The reconstruction of pre- and post-failure surfaces is carefully done with GIS and serves as a reference to future studies (Figures 6 and 7).

Specific Comments

The estimation of the shear strength of the basal sliding zone is based on back-analyses with UDEC for a pre-defined rupture plane geometry and an orthogonal set of fully persistent fractures in the sliding mass with uniform geometric and geomechanical properties. It is known from many previous studies (for example Vajont), that the strength and stability of slides with compound ruptures also depends on the internal strength of the sliding mass. However, the study does not evaluate the effect of sliding mass properties on the back-calculated basal shear zone strength. In addition, the rock-mechanical modeling assumptions are poorly constrained and not related to field and lab data. In fact it seems that some of the field observations (for example the stepped rupture plane) are violated. I suggest to perform a modeling study which is less trivial, which considers the few available field data and known depth trends of fracture properties as good as possible. It might be fruitful, to treat the basal rupture plane not as a pre-defined fully developed shear zone, but to investigate how this surface formed progressively from a previously not fully interconnected fracture network. This would lead to more substantial insights into progressive failure, which is only superficially discussed in the current manuscript. Another comment refers to the discussion and analysis of the rock slide runout. It is known from previous studies (for example Aaron et al 2020, Frontiers in Earth Sciences 30), that the dynamic friction angle during rapid runout motion can differ substantially from the static friction angle required

to initialize the motion. With the new GIS models available in this study, it would be worthwhile, to review some of the dynamic property of the Köfels Rock Slide.

---

## Referee Comment (RC3) · Anonymous Referee #3 · 4 Nov 2020

GIS-based topographic reconstruction and geomechanical modelling of the Köfels Rock Slide

Zangerl, Schneeberger, Steiner and Mergili

The Köfels rock slide is one of the most well-known rapid landslides in the European Alps. It has fascinated researchers since the work of Erismann in 1977. In many ways this work duplicates the work of many previous researchers – including Erismann (mass balance, energy analysis, theory of frictional heating) and Brückl and Parodidis (finite element modelling). However, instead of applying continuum-type finite element models, the authors use a 2-D "discontinuum" approach using a DEM model.

In general the paper is well-written and well-structured and I believe that readers with a special interest in the Köfels slide will find the paper interesting and worth a short read. Although the paper addresses the speculation surrounding the Köfels event, it does not appear to resolve – with certainty – any of the issues concerning the release dynamics. This, I think, is to be expected and should not be judged too harshly. Note for example, the last sentence of the conclusions, "Additional triggering factors, for example impact of dynamic loading, have to be considered in further investigations. This line summarizes a section in the discussion (ll 435-450) basically advancing the idea that, "dynamic loading due to earthquake shaking was able to trigger tow of the largest rock slides in the Alps, located about 130km apart. The authors then go on with this idea, "…earthquake triggering … would have caused numerous (smaller) events…" and that further investigations comprising a variety of methods should be applied to resolve this hypothesis. The authors (thankfully) exclude climate induced triggers (permafrost warming, ll 425-435), as well as "reducing rock mass strength by rock mass fracturing and weakening …" (ll 414-425).

Thus, the paper in many ways centers around the "avalanche truism": Why did the slide release? Because the friction angle was low. Why was the friction angle low? Because the slide released. The real question is why, what mechanism led to this low angle? And can it be proven, with reasonable speculation, which is basically the science of rock slide geology. This is where the analysis of the modelling methods becomes important. Note the important sentence on line 424: "Based on the results of numerical modelling study it is inconceivable that slope failure occurred under pure static conditions …."

So, how did the authors model the release dynamics? Firstly, they used DEM methods "to model a thin and discrete basal sliding zone which is able to accumulate large shear displacements" (ll 221). Thus, the failure zone was introduced into the model and "…the main deformation within the system takes place through movement along discontinuities" (ll 241). The selected angle of friction varied between "20deg and 27deg" for scenario A and "25deg and 32deg" for scenario B (ll 260, Section 3.33). It is therefore hardly surprising that the authors obtain a friction angle of 24deg for A and friction angle of 28deg for B (see results in abstract lline 17-20). *To me this means that the results the authors obtain arise directly from the selected parameters.* In fact, I question whether a DEM or finite element model can supply other results, especially when the shearing is concentrated in weak shear zones. The elastic shear modulus of this zone was defined to be 22GPa (ll 262). This value is evidently a static value, valid for all time. Maximum modelled deformations in the shear zone are about 0.25m (l 339). The model, thus does not investigate the possibility of shear softening, it is excluded from the modelling a-priori. *Therefore, the conclusion the hypothesis that fragmentation or shear softening did not lead the triggering is questionable.*

The rock was considered to be discrete blocks with "contacts or interfaces". However, a "continuum mesh of finite difference zones defines the deformability of the rock mass" (ll230-235). Thus, the model description (for me) is somewhat confusing – was it a continuum model or a discontinuum model? It should be pointed out that modelling a shear interface is possible with standard finite element continuum codes. In my opinion the modelling of the deformability of the surrounding rock is important because it defines how any stress concentrations in the shear layer are carried (or "bridged") by the surrounding rock. *Here, I think additional figures are required showing exactly how the interface and rock-mass are modelled.* What are the continuum rock parameters and what are the block interface parameters. What are the deformations in the surrounding rock? Do the discrete blocks rotate, like a layer of ball-bearings? Do they slide? Are all parameters "linear elastic"? According to lines 239+, "Blocks are considered as linear elastic …", there is again no softening, or fragmentation in the surrounding rock. The figures in the paper showing stress distributions are misleading, because the stress concentrations are highly local – perhaps a "zoom-in" to the shear zone where stress concentrations exist, coupled with the bridging stress distributions in the rock massive, would help characterize the failure mechanism.

My final impression of the paper is that it is well-written, certainly of interest to the rock avalanche community. However, the paper applies circular arguments in the modelling – obtaining results that are directly defined by the input parameters and modelling assumptions. This gives the paper a highly speculative character, in which an external hypothesis (earthquake triggering) is advanced to hide the limitations of the modelling effort. From the text, I cannot see the advantages of the discrete element model over a standard finite element approach – especially because both involve adding a "weak" shear zone to the model. How this zone changes (softens, fragments, etc) over time is not considered by the authors, although I suspect the applied DEM model would allow more realistic material behavior.

I would recommend publication if the authors could take the standard linear elastic approach (presented in the is paper) and supplement it with a more complex rock-mechanical modelling, that would either substantiate, or refute, a shear softening hypothesis. Then, and only then, would it be possible to introduce the "dynamic triggering" hypothesis.

---

## Author Comment (AC1) · 14 Dec 2020

Dear Azealdeen Al-Jawadi, thank you very much for your interesting question. Due to the geological situation, the size and the specificity of the event, direct transferability of the results to neighbouring areas is probably not possible. Best regards Christian Zangerl

---

## Author Response (AR1)

**Authors response to Editor 17.05.2021**

Dear Editor,

herewith I would like to send you the revised version of the manuscript "*GIS-based topographic reconstruction and geomechanical modelling of the Köfels Rock Slide*". All comments to reviewers are added by blue coloured text.

Thank you very much and best regards

Christian Zangerl, on behalf of the co-authors

**Comments on Reviewer 1 - Gregor Ortner (Referee RC1)**

All in all, it is a good and solidly constructed paper that contains a lot of work and effort. The authors had an unusually good initial situation regarding the available data and comparative studies. They have taken good advantage of it. The used approach is reasonably chosen and corresponds to the common standards and practice in this field and leads to an interesting outcome. And thus contributes to an increase of knowledge in the field of land slide analysis.

Specific comments

Basically the modelling strategy is clear and reasonable. To be able to reconstruct and model a mass movements in retrospect, it is clear that some assumptions must be made. These assumptions have a major impact on the final result. As a reader I found it somewhat difficult to fully follow the choice of the assumptions and parameterization made for modelling. It would be desirable that the origin of a chosen parameters is clearly declared and referenced and the choice of the parameters is sufficiently justified. This concerns especially the chapter "3.3.2 Model geometry, boundary and initial conditions" and "3.3.3 Material properties" as well as and the choice of water horizons in the model-Model boundarys - Hydrostatic water pressure - Material properties.

Comment: All these recommendations will be considered in the revised manuscript by improving the structure and wording.

Revision by the authors: The manuscript has been extensively revised in accordance with reviewer's comments, with the modelling section completely rewritten to include new extensive and time-consuming modelling results based on UDEC.

The choice of geotechnical parameters as well as the groundwater horizons should be addressed in the discussion and covered in a separate sub-section "Uncertainties" or "Model Uncertainties". In this subsection also the results of the modeling (with and without pore water pressure) should be discussed and critically reviewed.

Comment: It is intended to follow the recommendations of the reviewer by a comprehensive revision of the discussion chapter.

Revision by the authors: Done, see word document in correction mode, new modelling, new figures, new text.

Technical corrections

Line: 84: "exceptional high groundwater levels" -> It would be interesting to know how to reach this assumption Line:110 – 114:

Comment: We will describe this assumption in a detailed and traceable way.

Revision by the authors: Done

Mention when the landslide was.

Comment: We will include this information.

Revision by the authors: Done

Line:119: Briefly mention where the value of the volume 3km3 originates from.

Comment: A reference will be included here.

Revision by the authors: This information was already there.

Line:127: "Tauferberg" in Fig.1 not not labeled in Fig.2 called "Tauferer Berg" please label uniformly

Line:128: please label "Horlauchtal valley" in the figures

Comment: All figures will be checked again for correctness and improved if necessary.

Revision by the author: All figures have been reviewed, with Figure 8 to Figure 12 being completely redone.

Line:135-137: "This distinctive fragmentation of rock led to radon gas emissions and locally radioactive springs, which still affects today's population in Umhausen and causes noticably high cancer rates (Purtscheller et al. 1995)." -> interesting but irrelevant

Comment: We will delete this sentence in the revised manuscript.

Revision by the authors: Deleted

Line:158 - 160: "To what extent permafrost degradation is able to trigger a deep-seated rock slides characterized by a shear zone at a depth of several hundred metres, is unclear and still under discussion (Nicolussi et al. 2015)." -> perhaps not even worth mentioning.

Comment: We will delete this sentence in the revised manuscript.

Revision by the authors: Deleted

Line:215: "Estimation of rock mass strength and shear strength of discontinuities was done" -> please briefly explain how.

Comment: We will include additional descriptions to improve the readability and understandability.

Revision by the authors: Improved and rewritten

Figures: Style is all a bit "old school" but of course sufficient. Sometimes the labeling of Figs is not consistent with the text. Fig.1d: it would be nice if the jointsets are marked with colored lines and flags for the dip for better visualisation.

Comment: It is intended to check all figures with regard to information content and appearance and to adapt them if necessary.

Revision by the authors: Done, see above.

**Comments on Reviewer 2 - Anonymous (Referee RC2)**

Specific Comments

The estimation of the shear strength of the basal sliding zone is based on backanalyses with UDEC for a pre-defined rupture plane geometry and an orthogonal set of fully persistent fractures in the sliding mass with uniform geometric and geomechanical properties.

It is known from many previous studies (for example Vajont), that the strength and stability of slides with compound ruptures also depends on the internal strength of the sliding mass. However, the study does not evaluate the effect of sliding mass properties on the back-calculated basal shear zone strength.

Comment: We absolutely agree with the reviewer and therefore we will also focus on the comment in the revised manuscript. This will include a re-analyses of the existing numerical models, but if necessary new modelling studies based on UDEC to focus on the characteristics and influence of the internal rock mass deformation on stability behaviour. The new results will be included in the revised manuscript.

Revision by the authors: The manuscript has been extensively revised in accordance with reviewer's comments, with the modelling section completely rewritten to include new extensive and time-consuming modelling results based on UDEC. We performed more than 50 new UDEC models during the last months, focusing on the initial deformation and failure processes and back-calculation of the reconstructed basal shear zone. We also included discontinuity network mapped in the field. The new results were presented and discussed with recently published literature, especially with regard to earthquake triggering.

In addition, the rock-mechanical modeling assumptions are poorly constrained and not related to field and lab data. In fact it seems that some of the field observations (for example the stepped rupture plane) are violated.

Comment: The observed rupture surface of the Köfels Rock Slide is entirely located in competent granitic rock masses (Augengneiss). In our study we considered laboratory tests (UCS, Young's modulus, Poison ratio) from this type of rock, and performed a detailed field investigation campaign concerning the discontinuity network. Based on laboratory test and discontinuity date we applied rock mass classification systems (GSI) to estimate roughly the rock mass strength. We agree that this empirical approach has some limitations, and of course in-situ shear tests of the discontinuities

forming the basal rupture surface would be nice to have, but in our opinion hardly fundable. Most large-scale in-situ shear tests considering real stress conditions were performed in the scope of large dam projects (Please see Fishman 2014, Shear resistance along rock mass discontinuities: results of large-scale field tests, 41:6,1029-1034.). Nevertheless, it is assumed that the required rock mass properties can be estimated reasonably. In this context we would like to mention that the primary objective of the numerical modelling study was the back-calculation of the fully persistent basal shear zone which was formed by progressive failure during initial failure process.

In order to improve the manuscript and consider reviewer's suggestion we will improve, rewrite and change section 3.3.3 Material properties i.e. the input for the numerical modelling study, also by considering section 4.2.

Revision by the authors: All this has been done – please check manuscript in correction mode.

I suggest to perform a modeling study which is less trivial, which considers the few available field data and known depth trends of fracture properties as good as possible. It might be fruitful, to treat the basal rupture plane not as a pre-defined fully developed shear zone, but to investigate how this surface formed progressively from a previously not fully interconnected fracture network. This would lead to more substantial insights into progressive failure, which is only superficially discussed in the current manuscript.

Comment: This suggestion is scientifically interesting and understandable, and gets to the heart of the initial formation of any landslide. Understanding progressive failure in fractured rock masses is the key in landslide research, and for us it is highly obvious that progressive failure is the main mechanism of the failure process at the Köfels rock slide. Progressive failure is related to a complex interaction between existing natural discontinuities and brittle fracture propagation through intact rock bridges which finally leads to the formation of fully persistent basal shear zone. At this stage the slope is able to fail. Furthermore, progressive failure is characterised by complex in-situ stress conditions (stress concentration) and time dependent strength degradation mechanisms which must also occur within the rock mass driving the slope towards instability.

Several numerical approaches to study the initial formation process due to progressive failure of a rock slide are available. For example, Brueckl and Parotidis (2001, 2005) performed 2D FEM calculations (continuum) to study the development of the "creeping rock mass", which represents the initial phase of the Köfels rock slide. In this model a transition of the originally compact rock mass to ''soft'' rock, controlled by a Mohr–Coulomb and no tension yield criterion was assumed. These FEM models focus on the initial failure process by considering progressive failure mechanisms, at least in a simplified way. Pre-existing discontinuities and therefore the anisotropic nature of the rock mass or the failure geometry was not considered. Furthermore, in order to study the progressive failure mechanism as realistic as possible, discontinuities and (time-dependent) fracture mechanics need to be considered. This can be done, for example, by the software ELFEN, which is able to simulate crack growth and coalescence. Applying ELFEN (or similar software products able to model fracture mechanics) to the Köfels rock slide would be highly interesting, but can be seen as a comprehensive standalone study for future investigations.

The scope of our study was to back-calculate the shear strength properties of the fully persistent basal rupture surface which developed by progressive failure and fracture coalescence. As a hypothesis for our study, ongoing slope deformation is only possible if the shear strength properties of the basal shear zone are sufficiently low to allow slip and subsequently acceleration to high velocities (i.e. transition to the dynamic friction angle). By implementing the reconstructed - but as realistic as possible - slope geometry and basal rupture surface into a distinct (discrete) element model provide slope scenarios

(also different groundwater conditions) with a strong control on the "real" failure geometry and the shear strength parameters.

In our revised manuscript we intend to describe the working hypotheses and research questions in more detail (also the advantage of distinct element models) and include an extensive discussion of the concept of progressive failure in relationship to the Kofels Rock slide. In our revised manuscript we plan to improve the distinct element modelling part. Since we see certain advantages in using the distinct element method and a good complement to existing studies, we have not planned to use other software products, for example based on a continuum approach.

Revision by the authors: This has been done. Especially the numerical modelling part has been completely revised and a new model campaign focusing on possible initial failure mechanisms has been performed. Results are presented in the revised version - please check manuscript in correction mode.

Another comment refers to the discussion and analysis of the rock slide runout. It is known from previous studies (for example Aaron et al 2020, Frontiers in Earth Sciences 30), that the dynamic friction angle during rapid runout motion can differ substantially from the static friction angle required.

Comment: Dynamic motion is a very important aspect of large, rapid landslide processes such as the Köfels event. We agree with the referee that the friction angles relevant for dynamic motion are completely different than those for landslide release – an experience to be made when optimizing the parameters for running mass flow models such as RAMMS or r.avaflow. Köfels represents even more a specific case in this respect, as melting processes have occurred at the sliding surface, a rare phenomenon documented only for very few cases worldwide.

We would like to include an analysis of dynamic motion, but to our knowledge, there is some lack in software products available to date which is able to appropriately reproduce large, rapid sliding processes where the rock slide mass is internally deforming to some extent, but not displaying a flow-like behaviour. We have tried various times to back-calculate the Köfels event with the r.avaflow software, but as it was to be expected with a mass flow simulation tool, the derived deformation and lateral spreading was much too strong, compared to the observation, so that we have decided not to include the result in the paper. There is certainly a need for simulation models accounting for this type of motion – an aspect we will emphasize more strongly in the discussion of the revised manuscript.

Revision by the authors: A dynamic run out modelling was not done because we think this would be out of the scope of this article. However, such modelling study would be very interesting and should be done in the future in the framework of an own article.

**Comments on Reviewer 3 - Anonymous (Referee RC3)**

Thus, the paper in many ways centers around the "avalanche truism": Why did the slide release? Because the friction angle was low. Why was the friction angle low? Because the slide released. The real question is why, what mechanism led to this low angle? And can it be proven, with reasonable speculation, which is basically the science of rock slide geology. This is where the analysis of the modelling methods becomes important. Note the important sentence on line 424: "Based on the

results of numerical modelling study it is inconceivable that slope failure occurred under pure static conditions…

Comment: Our intention was not to simulate the progressive failure process characterised by fracture growth and coalescence (for additional information please see comments to Reviewer 2), but to determine the shear strength properties of a fully persistent basal rupture surface when the progressive failure process is completed. But, we fully agree that the question for the mechanism is most important and progressive failure is the main initial failure mechanism. As this is a study investigating an event that occurred around 9,500 years ago, the possible trigger(s) is (are) unknown and thus can only be reconstructed by indirect methods (e.g. lake-sediment based paleoseismology, climatic records). We choose the way to combine (i) estimations of critical values of rock mechanical parameters, and (ii) computer simulations to investigate which parameter values would have been necessary to allow failure under given groundwater scenarios, and (iii) to match the two parts. This is exactly what we did, and for example we found that extreme groundwater conditions may not have triggered the event. In the revised manuscript, we will try to more clearly explain this important aspect of our work.

Revision by the authors: The manuscript has been extensively revised in accordance with reviewer's comments, with the modelling section completely rewritten to include new extensive and time-consuming modelling results based on UDEC. We included now a modelling study focusing on the initial deformation and failure processes. These UDEC model are based on the main outcomes of discontinuity mapping in the field and on the elasto-plastic behaviour of the rock blocks by implementing a Mohr-Coulomb constitutive model. So, these models consider the concept of intact block failure and progressive failure, at least to some extent.

So, how did the authors model the release dynamics? Firstly, they used DEM methods "to model a thin and discrete basal sliding zone which is able to accumulate large shear displacements" (ll 221). Thus, the failure zone was introduced into the model and "…the main deformation within the system takes place through movement along discontinuities" (ll 241). The selected angle of friction varied between "20deg and 27deg" for scenario A and "25deg and 32deg" for scenario B (ll 260, Section 3.33). It is therefore hardly surprising that the authors obtain a friction angle of 24deg for A and friction angle of 28deg for B (see results in abstract lline 17-20). To me this means that the results the authors obtain arise directly from the selected parameters.

Comment: This is not the case, because we lowered the basal shear strength properties step-wise to reach slope failure conditions (i.e. transition from very small displacements with equilibrium to large displacements where equilibrium can no longer be achieved). Thus the obtained critical friction angles are the results from back-calculations with and without water pressures.

Revision by the authors: To be more understandable, we improved the structure, the figures and the text of the manuscript. Please see manuscript in correction mode.

In fact, I question whether a DEM or finite element model can supply other results, especially when the shearing is concentrated in weak shear zones. The elastic shear modulus of this zone was defined to be 22GPa (ll 262). This value is evidently a static value, valid for all time. Maximum modelled deformations in the shear zone are about 0.25m (l 339). The model, thus does not investigate the possibility of shear softening, it is excluded from the modelling a-priori. Therefore, the conclusion the hypothesis that fragmentation or shear softening did not lead the triggering is questionable.

The rock was considered to be discrete blocks with "contacts or interfaces". However, a "continuum mesh of finite difference zones defines the deformability of the rock mass" (ll230-235). Thus, the model description (for me) is somewhat confusing – was it a continuum model or a discontinuum model? It should be pointed out that modelling a shear interface is possible with standard finite element continuum codes. In my opinion the modelling of the deformability of the surrounding rock is important because it defines how any stress concentrations in the shear layer are carried (or "bridged") by the surrounding rock. Here, I think additional figures are required showing exactly how the interface and rock-mass are modelled. What are the continuum rock parameters and what are the block interface parameters. What are the deformations in the surrounding rock? Do the discrete blocks rotate, like a layer of ball-bearings? Do they slide? Are all parameters "linear elastic"? According to lines 239+, "Blocks are considered as linear elastic …", there is again no softening, or fragmentation in the surrounding rock. The figures in the paper showing stress distributions are misleading, because the stress concentrations are highly local – perhaps a "zoom-in" to the shear zone where stress concentrations exist, coupled with the bridging stress distributions in the rock massive, would help characterize the failure mechanism.

Comment: Here, we have the impression that our description was unclear and partly misleading. For our revised manuscript we plan to improve the description of the modelling study considerably, including both input parameters and outcomes. We plan to restructure the modelling chapter and will address all the questions and comments mentioned above. Among other aspects, this include for example a clear distinction between constitutive laws and parameters of blocks and structures (i.e. interfaces between blocks).

Revision by the authors: This has been done - please see manuscript in correction mode.

My final impression of the paper is that it is well-written, certainly of interest to the rock avalanche community. However, the paper applies circular arguments in the modelling – obtaining results that are directly defined by the input parameters and modelling assumptions. This gives the paper a highly speculative character, in which an external hypothesis (earthquake triggering) is advanced to hide the limitations of the modelling effort. From the text, I cannot see the advantages of the discrete element model over a standard finite element approach – especially because both involve adding a "weak" shear zone to the model. How this zone changes (softens, fragments, etc) over time is not considered by the authors, although I suspect the applied DEM model would allow more realistic material behavior.

Comment: Please see also comments to reviewer 2. The primary goal of this study was the back-calculation of the shear strength properties of a fully-persistent basal shear zone. The advantage of applying the distinct element model opposed to classical continuum approaches is, that the "real" geometries of the rock slide can be implemented. Furthermore, we know from other case studies that the active parts of basal rupture or shear zones are very narrow compared to the thickness of entire rock slide. And such localised displacements can be modelled quite well with the distinct element method, especially if larger shear offsets are occurring.

In addition, we think that the continuum approach including some type of strain softening consideration (i.e. progressive failure) is already quite well covered by the work of Brueckl and Parotidis (2001, 2005). They performed 2D FEM calculations (continuum) to study the development of the "creeping rock mass", which represents the initial phase of the Köfels rock slide. In this model a transition of the originally compact rock mass to ''soft'' rock, controlled by a Mohr–Coulomb and no tension yield criterion was assumed. These FEM models focus on the initial failure process by considering progressive failure mechanisms, at least in a simplified way. But pre-existing

discontinuities and therefore the anisotropic nature of the rock mass or the failure geometry was not considered.

Thus, the application of the distinct element method in this study is considered to be a useful addition and continuation to explore selected questions and hypotheses. Knowing, of course, that the cause of the Köfels rock slide, and in particular the trigger factors, are still not fully understood.

Revision by the authors: We still focused on the application of the distinct element method, implemented into the code UDEC. But as criticized, we improved the modelling chapter considerable by performing a total new modelling campaign. We studied the initial failure process, the impact of discontinuities on the failure mechanisms, the impact of discontinuities on internal deformation of the rock slide, etc. All model scenarios were performed by considering dry and groundwater flow conditions. Please see manuscript in correction mode.

I would recommend publication if the authors could take the standard linear elastic approach (presented in the is paper) and supplement it with a more complex rock-mechanical modelling, that would either substantiate, or refute, a shear softening hypothesis. Then, and only then, would it be possible to introduce the "dynamic triggering" hypothesis.

Comment: Given that FEM calculations have already been done by Brueckl & Parotidis (2001, 2005), we prefer to remain in the manuscript with the distinct element method. Nevertheless, considerable improvements are planned, and if useful and necessary also new distinct element models. A more comprehensible formulation of the research questions, as well as a comprehensive revision of the discussion, will be sought.

Revision by the authors: As already mentioned above we included a new modelling study, focussing on the initial failure mechanisms. We also included an elasto-plastic constitutive model with a tensile failure cut-off for the simulation of block failure (in shear and tension mode). The approach is interesting, because we consider a combination of both elasto-plastic block failure and shear failure/displacement as well as opening along discontinuities. The hypothesis of earthquake triggering was not the scope of our quasi-static modelling approach. However, a currently published article by Oswald et al. 2021 showed a single heavy earthquake may not have triggered the Köfels rock slide, but could be responsible for earthquake driven fatigue processes in the rock slope. We included new references and discussed this issue in the discussion chapter. Please see manuscript in correction mode.

---

## Author Response (AR2)

**Authors response to Editor 03.07.2021**

Dear Editor,

herewith I would like to send you the revised version of the manuscript "GIS-based topographic reconstruction and geomechanical modelling of the Köfels Rock Slide".

All comments to reviewer #3 are added by red coloured text.

Thank you very much and best regards

Christian Zangerl, on behalf of the co-authors

**Comments on Reviewer 3 - Anonymous (Referee RC3)**

I enjoyed reading the second revision of the paper. I found the article to be well written and informative. I especially liked the idea that modern remote sensing techniques (and DEM) can be used to "revisit" the major rockslides. Indeed, much progress has been made in thiese areas.

Some minor points:

Line 160 Instead of 1.6(10^16) J, maybe write 1.6(10^7) GJ. Maybe GJ is a better unit.
Comment: Changed by the authors

Line 280 I read the recent Science article concerning the Chamoli rock/ice avalanche. There the authors reported the volume to be about 27 mio m3. The Piz Cengalo slide was approx 2 mio m3. Here the volumes are reported in km3 = 10^9 m3. Therefore the volume is 3100 mio m3? This is immense. Perhaps the authors should write 3.1 km3 = 3100 mio m3 to give the reader an impression of the volume.
Comment: Changed by the authors

Discussion of the DEM method. There are problems with the DEM method. For example, all the stresses are based on an accurate calculation of the elastic strain. As long as the material remains "in place" and deformations are small this is not a problem. However, when deformations become large, the calculated state of strain/stress are approximations at best.
Comment: The authors fully agree that the DEM method has its limitations and DEM models can only address selected research questions. Due to the extreme challenges in determination the boundary conditions and rock mass properties as well as assigning appropriate rock mechanical material laws for such a large-scale rock slide event, numerical modelling – and this case the DEM method – can only provide approximations and insights in possible failure and deformation processes.

The qualities of the figures is excellent. Very clear.
Comment: No changes needed

I wonder about the conclusions. The last line of the paper is insightful: "Thus, a particular geological disposition ... responsible for the ... slide" That is, we were unable to model the slide with resonable parameters, therefore we must search for other triggers --seismic, progressive strength degradation ... etc. Couldn't there be another conclusion. There is something wrong with the model? Something is

missing. If other researchers will confront this problem again (they certainly will), what modelling advice would you provide? Or, are the boundary conditions/material parameters too unknown to improve the modelling?

Comment: Additional explanations were attached to the manuscript.

Since DEM modelling can only consider some aspects of the expected rock mechanical processes, it makes sense to carry out additional numerical modelling studies based on other approaches and loading conditions. In addition, subsurface investigations and rock mechanics tests would help to better determine the parameters of the rock and the discontinuities as well as the structure of the rock mass. Due to the exceptionally large volume of the rock slide, it should also be examined whether the classical laws of rock mechanics can fully represent the natural event.